# Local frustration determines loop opening during the catalytic cycle of an oxidoreductase

Lukas S Stelzl[1†], Despoina AI Mavridou[2], Emmanuel Saridakis[3], Diego Gonzalez[4], Andrew J Baldwin[5], Stuart J Ferguson[1], Mark SP Sansom[1]*, Christina Redfield[1]*

[1]Department of Biochemistry, University of Oxford, Oxford, United Kingdom; [2]Department of Molecular Biosciences, University of Texas at Austin, Austin, United States; [3]Institute of Nanoscience and Nanotechnology, NCSR Demokritos, Athens, Greece; [4]Laboratoire de Microbiologie, Institut de Biologie, Université de Neuchâtel, Neuchâtel, Switzerland; [5]Department of Chemistry, Physical and Theoretical Chemistry Laboratory, University of Oxford, Oxford, United Kingdom

**Abstract** Local structural frustration, the existence of mutually exclusive competing interactions, may explain why some proteins are dynamic while others are rigid. Frustration is thought to underpin biomolecular recognition and the flexibility of protein-binding sites. Here, we show how a small chemical modification, the oxidation of two cysteine thiols to a disulfide bond, during the catalytic cycle of the N-terminal domain of the key bacterial oxidoreductase DsbD (nDsbD), introduces frustration ultimately influencing protein function. In oxidized nDsbD, local frustration disrupts the packing of the protective cap-loop region against the active site allowing loop opening. By contrast, in reduced nDsbD the cap loop is rigid, always protecting the active-site thiols from the oxidizing environment of the periplasm. Our results point toward an intricate coupling between the dynamics of the active-site cysteines and of the cap loop which modulates the association reactions of nDsbD with its partners resulting in optimized protein function.

**\*For correspondence:**
mark.sansom@bioch.ox.ac.uk (MSPS);
christina.redfield@bioch.ox.ac.uk (CR)

**Present address:** †Department of Theoretical Biophysics, Max Planck Institute of Biophysics, Frankfurt, Germany

**Competing interests:** The authors declare that no competing interests exist.

## Introduction

Molecular recognition, the specific non-covalent interaction between two or more molecules, underpins all biological processes. During protein-protein association, molecular recognition often depends on conformational changes in one or both of the protein partners. Important insight into the mechanisms of molecular recognition has come from NMR experiments probing dynamics on a range of timescales (*Baldwin and Kay, 2009*; *Sekhar et al., 2018*). NMR, and especially the relaxation dispersion method, is unique in providing not only kinetic, but also structural information about the conformational changes leading up to protein association (*Lange et al., 2008*; *Sugase et al., 2007*). In some cases, molecular dynamics (MD) simulations have been successfully paired with NMR experiments (*Olsson and Noé, 2017*; *Pan et al., 2019*; *Robustelli et al., 2012*; *Zhang et al., 2016*; *Wang et al., 2016*; *Stiller et al., 2019*) to provide a more complete picture of the conformational dynamics landscape (*Paul et al., 2017*; *Peters and de Groot, 2012*; *Robustelli et al., 2013*). However, we still do not have a general understanding of the molecular determinants that enable some proteins to access their bound-like conformation before encountering their binding partners in a 'conformational-selection' mechanism, while others undergo conformational change only after recognizing their ligands in an 'induced-fit' mechanism (*Boehr et al., 2009*).

Local structural frustration (*Baldwin et al., 2011b*; *Ferreiro et al., 2018*; *Freiberger et al., 2019*), the existence of multiple favorable interactions which cannot be satisfied at the same time, has emerged as an important concept in rationalizing why some regions in proteins undergo

conformational changes in the absence of a binding partner (*Ferreiro et al., 2007*; *Ferreiro et al., 2011*; *Ferreiro et al., 2014*). In general, globular proteins are overall minimally frustrated and, thus, adopt well-defined native structures. Nonetheless, the existence of a set of competing local interactions can establish a dynamic equilibrium between distinct conformational states. Locally frustrated interactions (*Jenik et al., 2012*) are indeed more common for residues located in binding interfaces of protein complexes (*Ferreiro et al., 2007*) and can play an important role in determining which parts of these proteins are flexible. However, very few studies (*Danielsson et al., 2013*; *Das and Plotkin, 2013*) have analyzed in atomic detail how frustration shapes conformational dynamics and, therefore, can enable protein association. This lack of understanding hampers the exploitation of local frustration, and associated dynamic equilibria, as a design principle for generating optimal protein constructs that could be used in the production of synthetic molecular machines.

To investigate the role of frustration in conformational dynamics, we have used as a model system the N-terminal domain of the transmembrane protein DsbD (nDsbD), a key oxidoreductase in the periplasm of Gram-negative bacteria. DsbD is a three-domain membrane protein (*Figure 1A*) which acquires two electrons from cytoplasmic thioredoxin and, through a series of thiol-disulfide exchange reactions involving pairs of conserved cysteine residues, transfers these electrons to multiple periplasmic pathways (*Figure 1B/C*; *Cho and Collet, 2013*; *Depuydt et al., 2009*). nDsbD, the last component of the DsbD thiol-disulfide cascade, is essentially the redox interaction hub in the bacterial periplasm; it first interacts with the C-terminal domain of DsbD (cDsbD) to acquire the electrons provided by cytoplasmic thioredoxin and then passes these on to several globular periplasmic proteins, including DsbC, CcmG and DsbG (*Figure 1A/B*; *Stirnimann et al., 2006*). All these thiol-disulfide exchange steps involve close interactions between nDsbD and its binding partners, which are essential so that their functional cysteine pairs can come into proximity leading to electron transfer (*Haebel et al., 2002*; *Mavridou et al., 2011*). As nDsbD is essentially a reductase in the oxidative environment of the bacterial periplasm, its interactions with its multiple partners must happen in an optimal manner, making this protein an ideal model system to study.

X-ray structures show that the cap-loop region of nDsbD, containing residues D68-E69-F70-Y71-G72, plays a key role in the association reactions of this domain with its binding partners (*Haebel et al., 2002*; *Rozhkova et al., 2004*; *Stirnimann et al., 2005*). The catalytic subdomain of nDsbD is inserted at one end of its immunoglobulin (Ig) fold with the two active-site cysteines, C103 and C109, located on opposite strands of a β-hairpin (*Figure 1D*). For both oxidized and reduced nDsbD (with and without a C103-C109 disulfide bond, respectively), the active-site cysteines are shielded by the cap-loop region which adopts a closed conformation (*Figure 1D*; *Haebel et al., 2002*; *Mavridou et al., 2011*; *Goulding et al., 2002*). Strikingly, all structures of nDsbD in complex with its binding partners, cDsbD, DsbC, and CcmG (*Haebel et al., 2002*; *Rozhkova et al., 2004*; *Stirnimann et al., 2005*), show the cap loop in an open conformation that allows interaction between the cysteine pairs of the two proteins (*Figure 1D*; *Haebel et al., 2002*; *Stirnimann et al., 2005*; *Rozhkova and Glockshuber, 2007*).

While these X-ray structures provide static snapshots of the closed and open states of the cap loop, they do not offer any insight into the mechanism of loop opening that is essential for the function of nDsbD. To understand the drivers of this conformational change, two hypotheses regarding the solution state of nDsbD need to be examined (*Figure 1D*). Firstly, it is possible that the cap loop only opens when nDsbD encounters its binding partners, in an 'induced-fit' mechanism consistent with the hypothesized protective role of this region of the protein (*Goulding et al., 2002*). Secondly, it is also plausible that in solution the cap loop is flexible so that it samples open states, and thus allows binding of partner proteins via 'conformational selection' (*Boehr et al., 2009*). Of course, if the cap loop did open in the absence of a binding partner, that could compromise the transfer of electrons in an environment as oxidizing as the periplasm, and particularly through the action of the strong oxidase DsbA (*Denoncin and Collet, 2013*). Therefore, one could hypothesize that a protective role for the cap loop would be more important in reduced nDsbD (nDsbD$_{red}$), raising the question of whether nDsbD interacts with its partners via different loop-opening mechanisms depending on its oxidation state. In this scenario, the pair of cysteines could modulate the conformational ensemble of nDsbD as has been hinted at by structural bioinformatics (*Schmidt et al., 2006*; *Zhou et al., 2014*).

Here, we use both state-of-the-art NMR experiments and MD simulations to describe different behaviors for the cap loop depending on the oxidation state of nDsbD. The atomic-scale insight

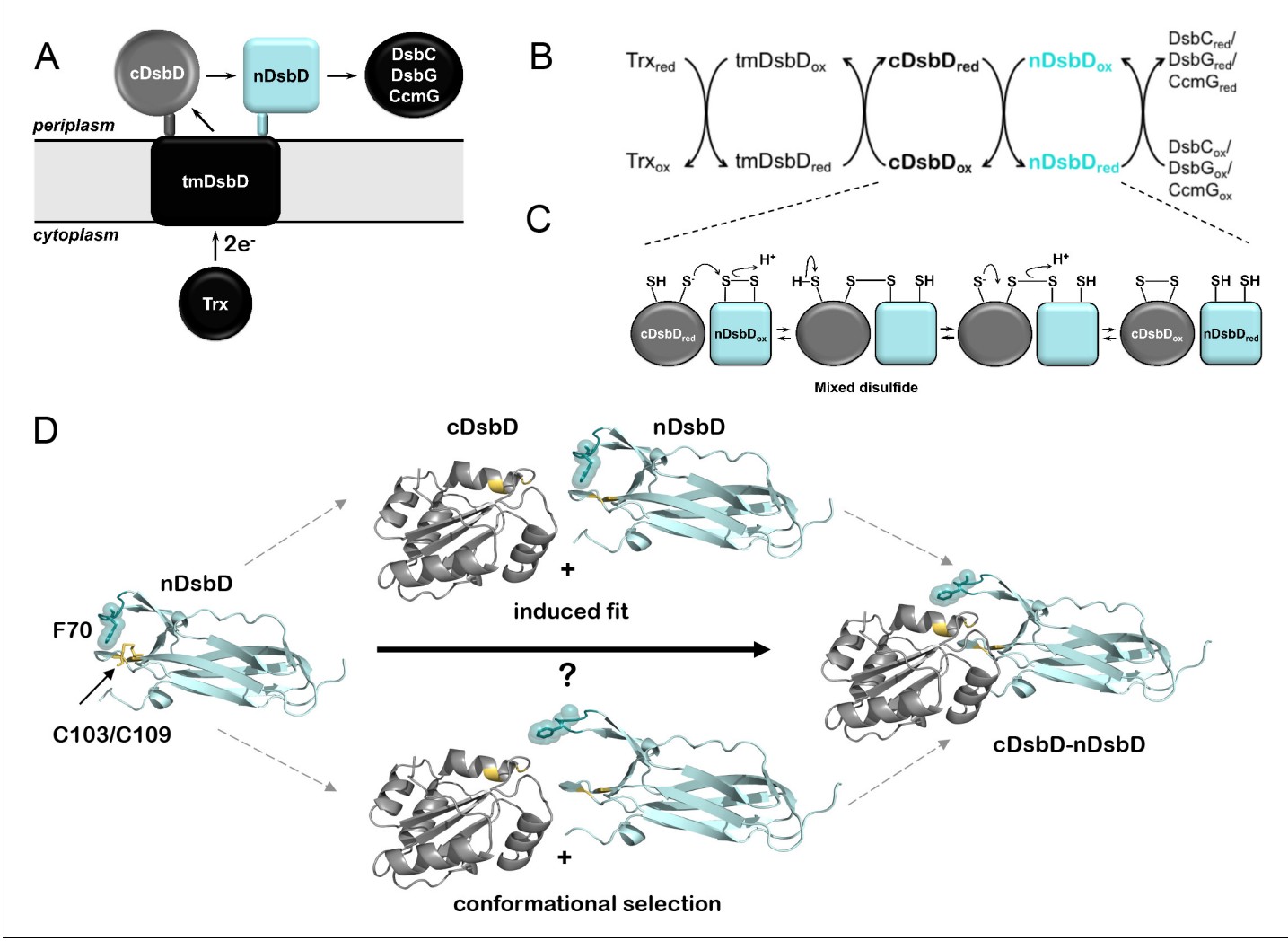

**Figure 1.** Schematic representation of DsbD. (**A**) DsbD comprises a central domain with eight transmembrane helices (tmDsbD) and two periplasmic globular domains (nDsbD and cDsbD). The electron transfer pathway from thioredoxin (Trx) to periplasmic partner proteins is shown. cDsbD, DsbC, CcmG and DsbG have a thioredoxin fold (circle), typical of thiol-disulfide oxidoreductases, while nDsbD adopts an immunoglobulin fold (square). nDsbD and cDsbD are shown in cyan and grey, respectively. (**B**) Electron transfer reactions involving DsbD are shown. Two electrons and two H$^+$ are transferred in each step. Trx$_{red}$ reduces the disulfide bond in tmDsbD$_{ox}$, tmDsbD$_{red}$ then reduces cDsbD$_{ox}$, and cDsbD$_{red}$ subsequently reduces nDsbD$_{ox}$. Oxidized DsbC, CcmG and DsbG subsequently accept electrons from nDsbD$_{red}$. (**C**) Schematic representation of the thiol-disulfide exchange reaction (bimolecular nucleophilic substitution) between cDsbD and nDsbD. (**D**) The central panels illustrate two possible mechanisms mediating cap loop opening in the formation of the nDsbD-cDsbD complex. In the upper scheme, a steric clash upon binding induces cap loop opening. In the lower scheme, the sampling of an open conformation by nDsbD allows formation of the complex without steric clashes. nDsbD$_{ox}$ is illustrated on the left (PDB: 1L6P); the cap loop (residues 68–72), which covers the active site is in teal. F70 (stick and surface representation) shields the active-site disulfide bond (yellow) in this closed conformation. The nDsbD cap loop adopts an open conformation in the complex between nDsbD and cDsbD (PDB: 1VRS), shown on the right, allowing interaction of the cysteine residues of the two domains. All structures in this and other figures were rendered in Pymol (Version 2.3.5, Schrödinger, LLC) unless otherwise indicated.

afforded by these two methods reveals how the disulfide bond introduces local frustration specifically in oxidized nDsbD (nDsbD$_{ox}$), allowing the cap loop and active site in nDsbD$_{ox}$ to sample open bound-like conformations in the absence of a binding partner. Our observations have implications not only for the function of DsbD, but more broadly for the role of local frustration in ensuring optimized protein-protein interactions.

## Results and discussion

### The solution structure of nDsbD as probed by NMR

NMR experiments demonstrate that the cap loop has an important protective role in shielding the active-site cysteines of nDsbD. For nDsbD$_{ox}$, we showed previously that reduction of the disulfide bond by dithiothreitol (DTT) is very slow, with only 50% of nDsbD$_{ox}$ reduced after ~2 hr. This indicates that the cap loop shields C103 and C109 from the reducing agent (*Mavridou et al., 2011*). In addition, NMR spectra for nDsbD$_{red}$ collected between pH 6 and 11 show no change in cysteine Cβ chemical shifts. This indicates that the p$K_a$ of C109, the cysteine residue thought to initiate reductant transfer (*Rozhkova et al., 2004*), is above 11, thus making it unreactive in isolated nDsbD$_{red}$. This suggests that shielding of the active site by the cap loop in nDsbD$_{red}$ protects C103 and C109 from non-cognate reactions in the oxidizing environment of the periplasm.

The cap loop shields the active site by adopting a closed conformation as in the X-ray structures of oxidized (PDB: 1L6P and 1JPE) (*Haebel et al., 2002*; *Goulding et al., 2002*) and reduced (PDB: 3PFU) (*Mavridou et al., 2011*) nDsbD. These structures are very similar to each other with pairwise Cα RMSDs for residues 12 to 122 of less than 0.15 Å. To probe the solution structure of oxidized and reduced nDsbD, we measured $^1$H-$^{15}$N residual dipolar couplings (RDCs), which are sensitive reporters of protein structure (*Figure 2—figure supplement 1* and *Figure 2—figure supplement 1—source data 1*). Analysis of these data indicates that the orientation of the active-site/cap-loop residues relative to the core β-sandwich of both oxidized and reduced nDsbD in solution is well described by the X-ray structures in which the cap loop adopts a closed conformation. Nevertheless, the cap loop must open to expose the active site in order for nDsbD to carry out its biological function.

### NMR spin-relaxation experiments and model-free analysis

To investigate whether the cap loop is flexible in solution and whether the dynamic behavior of the loop differs in the two oxidation states, we studied the fast time-scale (ps-ns) dynamics of nDsbD using NMR relaxation experiments. $^{15}$N relaxation experiments, analyzed using the 'model-free' approach (*Lipari and Szabo, 1982a*; *Lipari and Szabo, 1982b*), showed that most of the protein backbone is relatively rigid. The majority of order parameters (S$^2$) for the backbone amide bonds are above 0.8 (*Figure 2*). The N- and C-terminal regions of nDsbD gave very low S$^2$ values and are clearly disordered in solution. Other residues with order parameters below 0.75 are located in loops or at the start or end of elements of secondary structure, but are not clustered in a specific region. For example, lower order parameters are observed for residues 57 and 58 in both redox states. These residues are located in a long 'loop' parallel to the core β-sandwich but not identified as a β-strand due to the absence of hydrogen bonds.

In both oxidation states, the S$^2$ values for the cap-loop region are similar to the rest of the folded protein. Thus, on average, the cap loop of nDsbD does not undergo large amplitude motions on a fast timescale, which means that the active-site cysteines remain shielded. However, close inspection of the $\{^1$H$\}$-$^{15}$N NOE values for the cap-loop residues in oxidized and reduced nDsbD does show a clear difference between the two oxidation states (*Figure 2—figure supplement 2* and *Figure 2—figure supplement 2—source data 1*). In nDsbD$_{ox}$, the residues between V64 and G72 have NOE values at or below 0.7, while in nDsbD$_{red}$ there is only one residue with an NOE value at or below 0.7. The consequence of this difference is that the majority of amino acids in this region of nDsbD$_{ox}$ require a more complex model (with the $\tau_e$ parameter ranging from ~50 to 350 ps) to obtain a satisfactory fit in the analysis, in contrast to the majority of residues in nDsbD$_{red}$ which can be fitted with the simpler S$^2$ only model (with a $\tau_e$ value of faster than ~10 ps). Therefore, although the amplitude of motions of cap-loop residues is limited and does not appear to differ very much between the two oxidation states, the timescale of the fast dynamics, as described by the approach of Lipari and Szabo, does differ.

### Molecular dynamics simulations

To evaluate the differences between the conformational dynamics of reduced and oxidized nDsbD with atomic resolution, molecular dynamics (MD) simulations were employed. We tracked the orientation of the cap loop relative to the active site in the generated MD trajectories, by calculating the

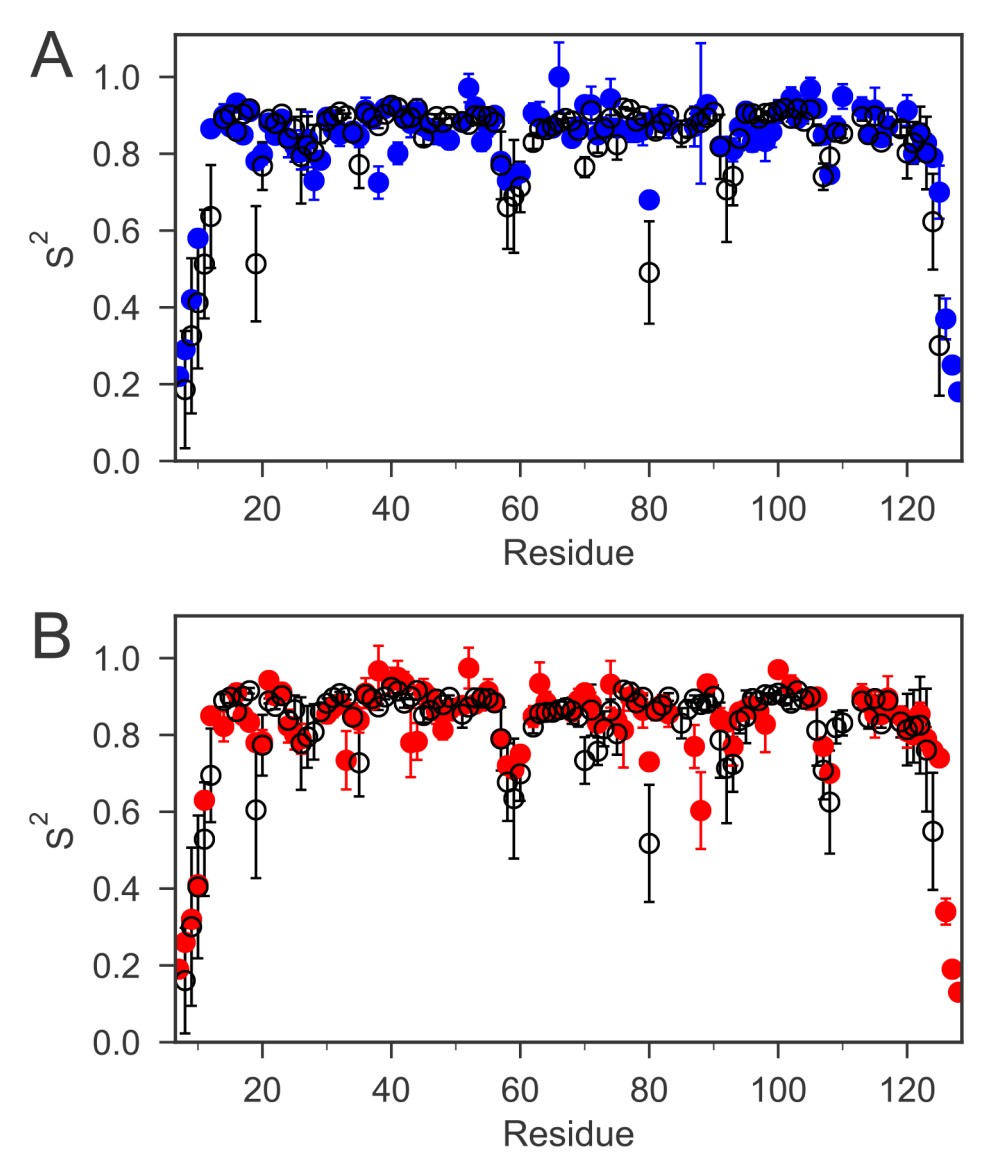

**Figure 2.** The order parameters ($S^2$) for backbone amides derived from $^{15}N$ relaxation experiments and MD simulations are plotted as a function of sequence. The experimental order parameters for reduced and oxidized nDsbD are shown in blue (**A**) and red (**B**), respectively. The average order parameters calculated from the combined 1 μs MD simulation ensembles starting from closed structures for reduced (**A**) and oxidized (**B**) nDsbD are shown in black. Errors in $S^2$ were determined by Monte Carlo simulations (experiment) and resampling of $S^2$ from trajectory segments (MD simulations). The simulations of both oxidized and reduced nDsbD reproduced even subtle features of sequence-dependent variation of $S^2$ values. For example, $S^2$ values for residues 57–60 were lowered in both experiment and simulation. These residues are not part of the regular secondary structure of the Ig fold in both the X-ray structures of nDsbD and in the MD simulations.

The online version of this article includes the following source data and figure supplement(s) for figure 2:

**Figure supplement 1.** $^1H - ^{15}N$ residual dipolar couplings (RDCs) for reduced and oxidized nDsbD.

**Figure supplement 1—source data 1.** Quality factors (Q) and alignment tensor parameters ($D_a$, R, θ, φ, ψ) obtained from fits of experimental RDCs to X-ray structures and from analysis of the MD trajectories.

**Figure supplement 2.** Fast time-scale dynamics of the cap loop.

**Figure supplement 2—source data 1.** Model-free analysis of $^{15}N$ relaxation data for the cap loop and the flanking β-strands.

**Figure supplement 3.** Comparison of backbone order parameters ($S^2$) derived from experimental $^{15}N$ relaxation data and calculated from the 1μs MD simulations for (**A**) reduced and (**B**) oxidized nDsbD.

distance between the center of the aromatic ring of F70 and the sulfur atom of C109. Multiple simulations, for a total of 1 µs, were started from the X-ray structure of reduced nDsbD (3PFU). The distance between F70 and C109 remained close to the value in the X-ray structure of nDsbD$_{red}$, as shown for sample trajectories in *Figure 3A and B* (*Figure 3—figure supplement 1*). In addition, C103 and C109 remained largely shielded, as assessed by the solvent accessible surface area (SASA) (*Figure 3—figure supplement 2*). Therefore, opening of the cap loop in nDsbD$_{red}$ was not observed in 1 µs of simulation time (*Figure 3D* and *Figure 3—figure supplement 1*). This confirms that the cap loop protects the active-site cysteines of nDsbD$_{red}$ from the oxidizing environment of the periplasm.

By contrast, nDsbD$_{ox}$ showed more complex dynamics in the 1 µs of simulations launched from the closed 1L6P X-ray structure (*Figure 3* and *Figure 3—figure supplement 1*). In one 200-ns simulation the cap loop remained closed (*Figure 3E*), only showing occasional fluctuations. In a different simulation the fluctuations in the cap loop were much more pronounced (*Figure 3F*); after ~80 ns the cap loop opened, as judged by the F70-C109 distance, and stayed open for the remainder of the 200 ns MD trajectory. A further 600 ns of simulations showed cap loop opening events with durations ranging from <1 ns to 22 ns (*Figure 3—figure supplement 1*, *Figure 5—figure supplement 1*). Calculations of SASA confirmed that opening of the loop increases the solvent exposure of C103 and C109 (*Figure 3—figure supplement 2*). The spontaneous loop opening of nDsbD$_{ox}$ and the relative stability of partially open conformations observed in our MD simulations (*Figure 3H*), mark a clear difference in the conformational ensembles sampled by oxidized nDsbD, as compared to those sampled by the reduced protein (*Figure 3D,H*).

We have also employed MD to probe the behavior of the cap loop in simulations starting from an open conformation of nDsbD observed in the 1VRS X-ray structure of the nDsbD/cDsbD complex (*Rozhkova et al., 2004*). Five of the eight trajectories that were started from an open loop conformation for nDsbD$_{red}$, closed within 10 ns (*Figure 3C*). Similarly, simulations of nDsbD$_{ox}$, that were started from a fully open 1VRS oxidized structure, closed within 10 ns in 3 out of 10 trajectories (*Figure 3G*). Together, these simulations demonstrate that in both reduced and oxidized nDsbD the cap loop can close rapidly in the absence of a bound interaction partner.

To validate the MD simulations, we used the generated trajectories to predict $^1$H-$^{15}$N RDC and S$^2$ values and to compare these to our experimental values. RDCs provide information about the average orientation of peptide bonds and are therefore well-suited for comparison with MD simulations. For both nDsbD$_{red}$ and nDsbD$_{ox}$, the agreement between calculated and experimental RDCs improved for the 1 µs MD ensemble compared to the static X-ray structures (*Figure 2—figure supplement 1* and *Figure 2—figure supplement 1—source data 1*). In addition, the extent of the fast-time scale (ps-ns) dynamics observed experimentally by NMR is on the whole correctly reproduced by the MD simulations (*Figure 2* and *Figure 2—figure supplement 3*). The trend for the cap loop is consistent with experiment; the S$^2$ values for the cap-loop region for both oxidized and reduced nDsbD are on par with the rest of the well-ordered protein core (*Figure 2—figure supplement 3* and *Figure 2—figure supplement 2—source data 1*). The observed cap-loop opening in the MD simulations of nDsbD$_{ox}$ is too rare an event to lower the S$^2$ values in a significant way. Nonetheless, the lowered experimental {$^1$H}-$^{15}$N NOE values for nDsbD$_{ox}$, and the need to fit them using a model that includes a $\tau_e$ parameter, are consistent with the observation of loop opening events in the MD simulations for nDsbD$_{ox}$. The good level of agreement between experimental NMR parameters and values calculated from the MD trajectories provides confidence that the 1 µs simulations of oxidized and reduced nDsbD allow us to glean a realistic picture of the fast timescale dynamics of nDsbD with atomic resolution.

## NMR and MD simulations reveal local frustration in nDsbD$_{ox}$

The differences in the cap loop dynamics, with the loop being more flexible in nDsbD$_{ox}$ than in nDsbD$_{red}$, must be a direct effect of the oxidation state of the two cysteine residues in the active site of nDsbD. X-ray crystallography (*Mavridou et al., 2011*) and solution NMR show that the average structures of the two redox states are very similar, with the only major difference between the two states being the presence of a disulfide bond in the active site of nDsbD$_{ox}$ in lieu of a pair of thiol groups in nDsbD$_{red}$.

Analysis of the MD simulations shows that the disulfide bond introduces local structural frustration in nDsbD$_{ox}$. The side chains of C103 and C109, which form the disulfide bond, adopt gauche-

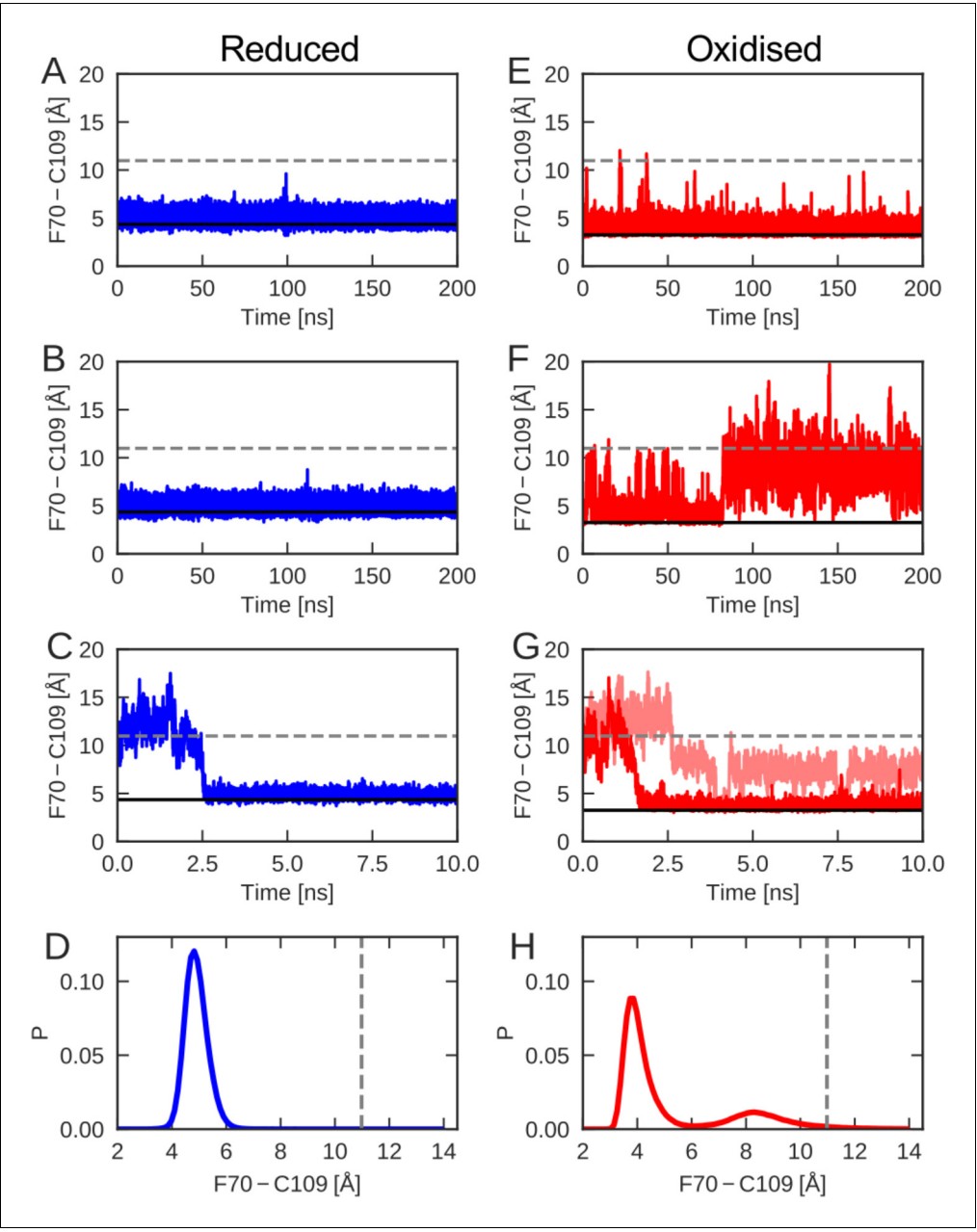

**Figure 3.** The cap-loop conformation was probed in MD simulations with reduced (**A–C**) and oxidized (**E–G**) nDsbD. The conformational state of the cap loop was followed by tracking the distance between the center of the aromatic ring of F70 and the sulfur atom of C109. The solid and dashed lines at ~3–4 and~11 Å show the F70 ring to C109 Sγ distance in the closed (3PFU/1L6P) and open (1VRS) X-ray structures, respectively. The cap loop remains stably closed in simulations of nDsbD$_{red}$, as shown in (**A**) and (**B**). The cap loop in nDsbD$_{red}$ closes within 3 ns for a simulation started from an open conformation based on the 1VRS structure (**C**). The 1 μs simulation ensemble for nDsbD$_{red}$ shows a closed conformation of F70 relative to C109 (**D**). The cap loop conformation is more flexible in simulations of nDsbD$_{ox}$; in some simulations it remains closed (**E**) while in others the loop opens (**F**). The loop closes in some simulations of nDsbD$_{ox}$ started from an open conformation (**G**). Overall, the 1 μs simulation ensemble for nDsbD$_{ox}$ shows a preference for the closed conformation of F70 relative to C109, with open conformations sampled relatively rarely (**H**).

The online version of this article includes the following figure supplement(s) for figure 3:

**Figure supplement 1.** Distances between the aromatic ring of F70 and the Sγ of C109 for additional MD simulations.

**Figure supplement 2.** Solvent accessible surface area (SASA) measured for MD simulations.

conformations ($\chi_1 \sim -60°$) in X-ray structures. This conformation is maintained throughout the MD simulations with only rare excursions (*Figure 4C*). The C103-C109 disulfide bond provides a flat binding surface for the side chain of F70. This aromatic ring packs onto the disulfide bond with its sidechain in either a gauche- or a trans orientation (*Figure 4A and B*). These two conformations are populated approximately equally and exchange rapidly in MD, on a timescale of tens of ns (*Figure 4—figure supplement 1*). This conformational averaging is evidenced experimentally by the almost identical F70 Hβ chemical shifts, which differ by only 0.015 ppm. TALOS-N (*Shen and Bax, 2013*) analysis of nDsbD$_{ox}$ chemical shifts does not predict a single fixed $\chi_1$ value for F70, consistent with the averaging between a gauche- and a trans orientation seen for this side chain in MD simulations.

By contrast, in nDsbD$_{red}$ F70 packs onto the active site in a single well-defined conformation, explaining why the cap loop is rigid in nDsbD$_{red}$. The side chains of C103 and C109 both adopt trans conformations ($\chi_1 \sim 180°$) in the X-ray structure of nDsbD$_{red}$. This conformation is maintained throughout the MD simulations of nDsbD$_{red}$ with only rare excursions (*Figure 4F*). This trans conformation of the cysteines, along with the fact that the two thiols are bulkier than a disulfide bond, result in the cysteine thiols providing a snug-fitting binding site for F70 (*Figure 4D*) in which one of the C109 Hβ interacts closely with the aromatic ring of F70. Consequently, F70 is locked in its gauche- conformation (*Figure 4E* and *Figure 4—figure supplement 1*). NMR data support this packing arrangement of the cap loop onto the active site of nDsbD$_{red}$. The Hβ of C109 are upfield shifted at 1.83 and 0.66 ppm, from a random coil value of ~3 ppm, consistent with the close proximity to the aromatic ring of F70. The two distinct Hβ shifts for F70 of nDsbD$_{red}$, which are separated by ~0.2 ppm, show NOEs of differing intensity from the backbone H$^N$, consistent with a single conformation for the F70 side chain. TALOS-N (*Shen and Bax, 2013*) analysis of chemical shifts predicts a gauche- $\chi_1$ conformation for F70, in agreement with the MD simulations.

Recently determined X-ray structures of nDsbD from *Neisseria meningitidis* (*Smith et al., 2018*) provide support for our observation of oxidation-state-dependent differences in the dynamics of the cap loop. Despite sharing only 27% sequence identity with the *E. coli* domain, nDsbD$_{red}$ from *N. meningitidis* shows a very similar structure with a closed 'Phen cap' loop in which the aromatic ring of F66 adopts a gauche- conformation packing tightly against C106. Interestingly, nDsbD$_{ox}$ from *N. meningitidis* crystallizes in the $P2_13$ space group with six molecules in the asymmetric unit. One of these molecules is observed to have an open 'Phen cap' loop which superimposes very well with the open conformation observed in our MD simulations (*Figure 4—figure supplement 2A*). The five molecules with a closed 'Phen cap' loop show F66 packing onto the C100-C106 disulfide bond in both trans and gauche- orientations reminiscent of the two almost equally populated closed structures observed for *E. coli* nDsbD$_{ox}$ in our MD simulations (*Figure 4—figure supplement 2B/C*). The range of 'Phen cap' loop conformations observed in *N. meningitidis* nDsbD$_{ox}$ are likely to reflect conformations sampled by the protein in solution. Thus, local structural frustration appears to be a conserved feature for nDsbD$_{ox}$ across many Gram-negative bacterial species.

## Structural features of cap loop opening

To understand how local structural frustration gives rise to cap loop opening in nDsbD$_{ox}$, we characterized loop opening events in our MD simulations. First, we investigated whether cap loop opening is linked directly to one of the two frustrated F70 $\chi_1$ conformers. The opening of nDsbD$_{ox}$ at 82.5 ns in *Figure 3F* coincides with a change of the F70 $\chi_1$ from gauche- ($-60°$) to trans ($-180°$) (*Figure 5* and *Figure 5—figure supplements 2–4*). During the 1 μs trajectory, the cap loop experiences several shorter opening events; twenty events with a duration of at least one ns and a F70-to-C109 distance of at least 10 Å are observed (*Figure 3*, *Figure 3—figure supplement 1*, *Figure 5—figure supplements 1*, *2*). In only one of these events does F70 undergo a similar gauche- to trans transition at the point of opening. Fourteen of the openings start from F70 in the trans conformer, while five openings start from the gauche- conformer. Thus, loop opening is not underpinned by a specific F70 $\chi_1$ conformer.

By contrast, we found that the dynamics of Y71 and the relative orientation of F70 and Y71 seem to be important for cap loop opening. In closed nDsbD$_{ox}$ Y71 also samples both gauche- and trans $\chi_1$ conformers, showing a strong preference for gauche- (86%). Despite this preferred conformation, we note that there is no correlation between the $\chi_1$ values of F70 and Y71; the less common Y71 trans conformer is observed for both frustrated F70 conformers. We observed the sequence of

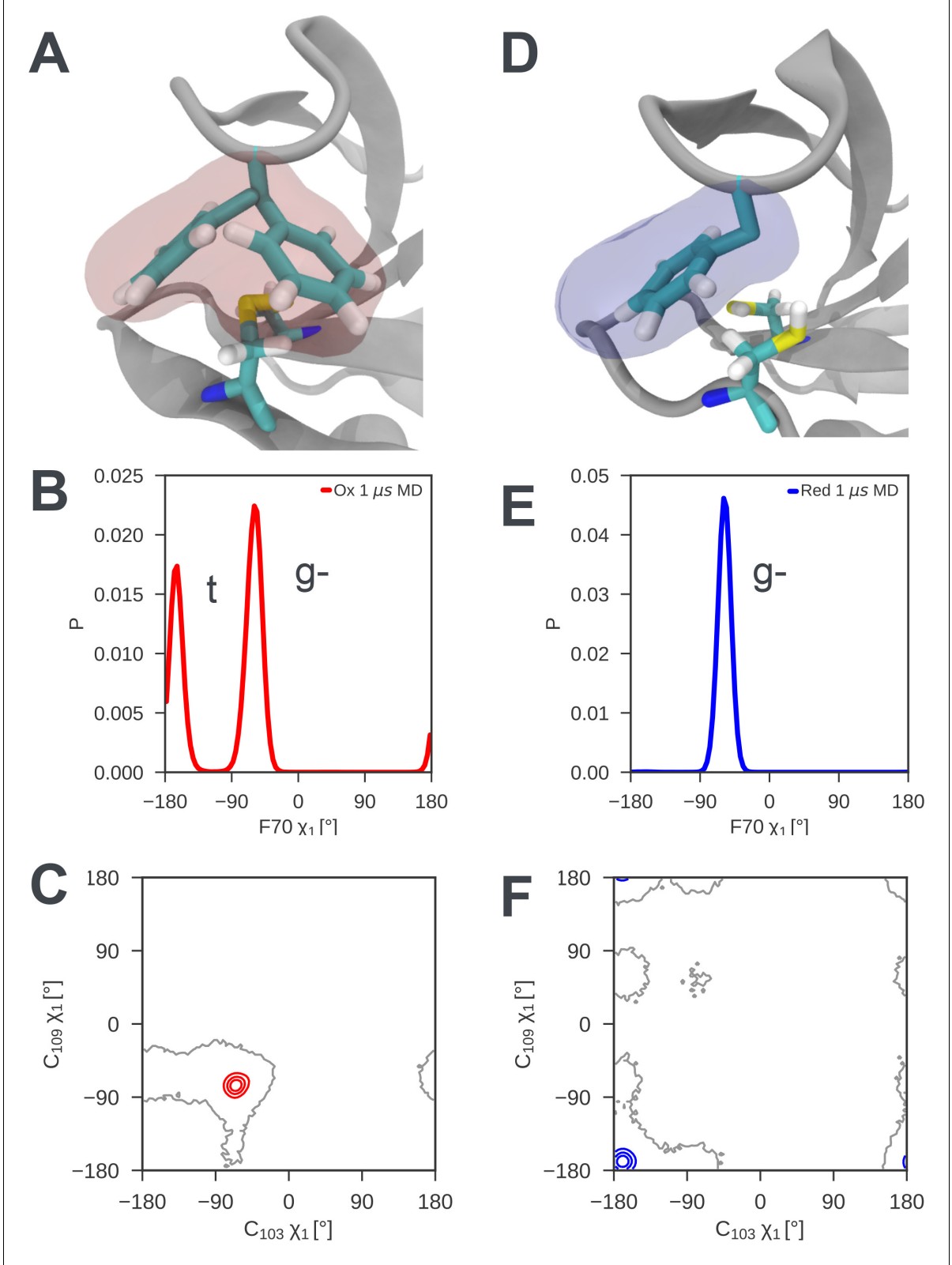

**Figure 4.** Local frustration destabilizes the cap loop of oxidized nDsbD while the cap loop of reduced nDsbD packs stably onto the active site. (A) In nDsbD$_{ox}$, the side chain of F70 switches between a gauche- and trans orientation in MD simulations as shown by the mass density calculated for F70 (red). (B) The gauche- and trans conformations are approximately equally populated in the 1 μs MD simulations of nDsbD$_{ox}$. (D) The cap loop and the active site of nDsbD$_{red}$ are rigid as shown by the mass density calculated for F70 (blue). The well-defined density closely circumscribes the position of

*Figure 4 continued on next page*

*Figure 4 continued*

the F70 side chain in the 3PFU crystal structure. (**E**) F70 is locked in the gauche- conformation. The surfaces in (**A**) and (**D**) surround points with at least 0.12 relative occupancy; these structures were rendered using VMD (*Humphrey et al., 1996*). The side chains of C103 and C109 in the active site of nDsbD adopt different conformations in the oxidized (**C**) and reduced (**F**) isoforms as shown by contour plots of their $\chi_1$ dihedral angles. Here 1 μs of MD simulations is analysed for each redox state.

The online version of this article includes the following figure supplement(s) for figure 4:

**Figure supplement 1.** F70 $\chi_1$ torsion angle during 1μs of MD simulations for reduced (left) and oxidized (right) nDsbD.

**Figure supplement 2.** Comparison of 'snapshots' of *E. coli* nDsbD$_{ox}$ from MD simulations with the X-ray structure of *Neisseria meningitidis* nDsbD$_{ox}$.

events leading to loop opening (*Figure 5* and *Figure 5—figure supplements 2–4*). Although the cap loop remains closed from 60 to 80 ns, the side chains of both F70 and Y71 sample a number of different conformations. From 60 to 65 ns, both F70 and Y71 are in the gauche- $\chi_1$ conformation observed in the 1L6P X-ray structure. At 65.9 ns, F70 $\chi_1$ switches to the trans conformer. This brings the F70 and Y71 rings closer together. From 66 to 80 ns, F70 remains primarily in the trans conformer with only brief excursions to gauche- (at ~72.7 and~77.7 ns), and at 79.9 ns F70 returns to the gauche- $\chi_1$ conformer. Meanwhile, at 76.9 and 80.3 ns the ring of Y71 transiently moves into more exposed conformations as a result of changes to $\chi_1$ and $\chi_2$, but the loop remains closed (*Figure 5* and *Figure 5—figure supplement 3*). Immediately before loop opening the F70 and Y71 rings are in close proximity, while Y71 is also close to C109. This flexible and dynamic nature of the side chains of F70 and Y71 in the closed state is reflected in their RMSD, to the equilibrated X-ray structure, observed during the MD trajectories. In the nDsbD$_{ox}$ 1 μs trajectories, F70 and Y71 have RMSDs of 2.8 ± 0.8 and 2.6 ± 0.7 Å, respectively when the loop is closed (F70-C109 < 6 Å). By contrast, these RMSD values are 1.7 ± 0.6 and 1.6 ± 0.8 for F70 and Y71, respectively in nDsbD$_{red}$ trajectories. It is worth noting here that the aromatic ring of F67 in *N. meningitidis* nDsbD$_{ox}$, which is homologous to Y71 in *E. coli* nDsbD, is also dynamic, adopting different conformations in the crystallized closed structures (*Figure 4—figure supplement 2*; *Smith et al., 2018*). Taking these observations together, we propose that the existence of local structural frustration for F70 results in increased dynamics for both F70 and Y71, which leads to occasional loop opening in nDsbD$_{ox}$.

Following opening of the cap loop at 82.5 ns, Y70 and Y71 continue to sample a variety of conformations, which could be important for the cap loop to access a fully open structure. The initial open states show the ring of F70 pointing inwards, due to its trans conformation. Transition of F70 to the gauche- conformer after loop opening, leads to a more extended orientation for F70. The extent of loop opening, as measured by the F70-to-C109 distance, fluctuates during the 120 ns open period; for example, at 85.6 ns the loop is in a more open conformation (12.5 Å) than at 93.2 ns (8.6 Å) (*Figure 5*). Interestingly, as the F70-to-C109 distance decreases, the loop is often prevented from closing fully by the side chain of Y71 which sits in between C109 and F70 in a 'blocking' conformation (for example at 93.2 ns and later at 151.9 and 155.8 ns [*Figure 5* and *Figure 5—figure supplement 5*]). The orientation of F70 relative to Y71 also changes while the loop is open. At 98.2 ns the position of Y70 is similar to that observed in the open 1VRS X-ray structure but Y71 points inwards. At 102.2 ns the Y70 and F71 orientations are more similar to the 1VRS structure (*Figure 5*). The cap loop can also adopt conformations that are more open than the open 1VRS structure (at ~145 and~181 ns); these structures are characterized by larger F70-C109 and Y71-C109 distances (>16 Å and 10 Å, respectively) (*Figure 5—figure supplements 2, 5*). These conformations may be important for the initial docking of cDsbD prior to nDsbD/cDsbD complex formation, as they would minimize the steric clashes between the two domains during binding.

The backbone $\phi$ and $\psi$ torsion angles of D68, E69, F70, Y71 and G72 also show clear changes at 82.5 ns when the loop begins to open (*Figure 5* and *Figure 5—figure supplement 4*). E69 $\phi$ and $\psi$ become less variable in the open loop while the $\phi$ and $\psi$ values for D68 and F70 become more variable. These torsion angle changes are correlated with loss of the hydrogen bond between the amide nitrogen of Y71 and the side chain carboxyl group of D68 upon loop opening. This hydrogen bond is observed in the 1L6P X-ray structure and in 95% of structures between 60 and 82.5 ns, prior to loop opening. Notably, D68 and E69 adopt $\phi$ and $\psi$ values more reminiscent of the closed state between 160 and 180 ns, despite the loop remaining open (*Figure 5—figure supplement 4*); during this period the Y71HN-D68Oδ hydrogen bond is again observed in 62% of structures. By contrast,

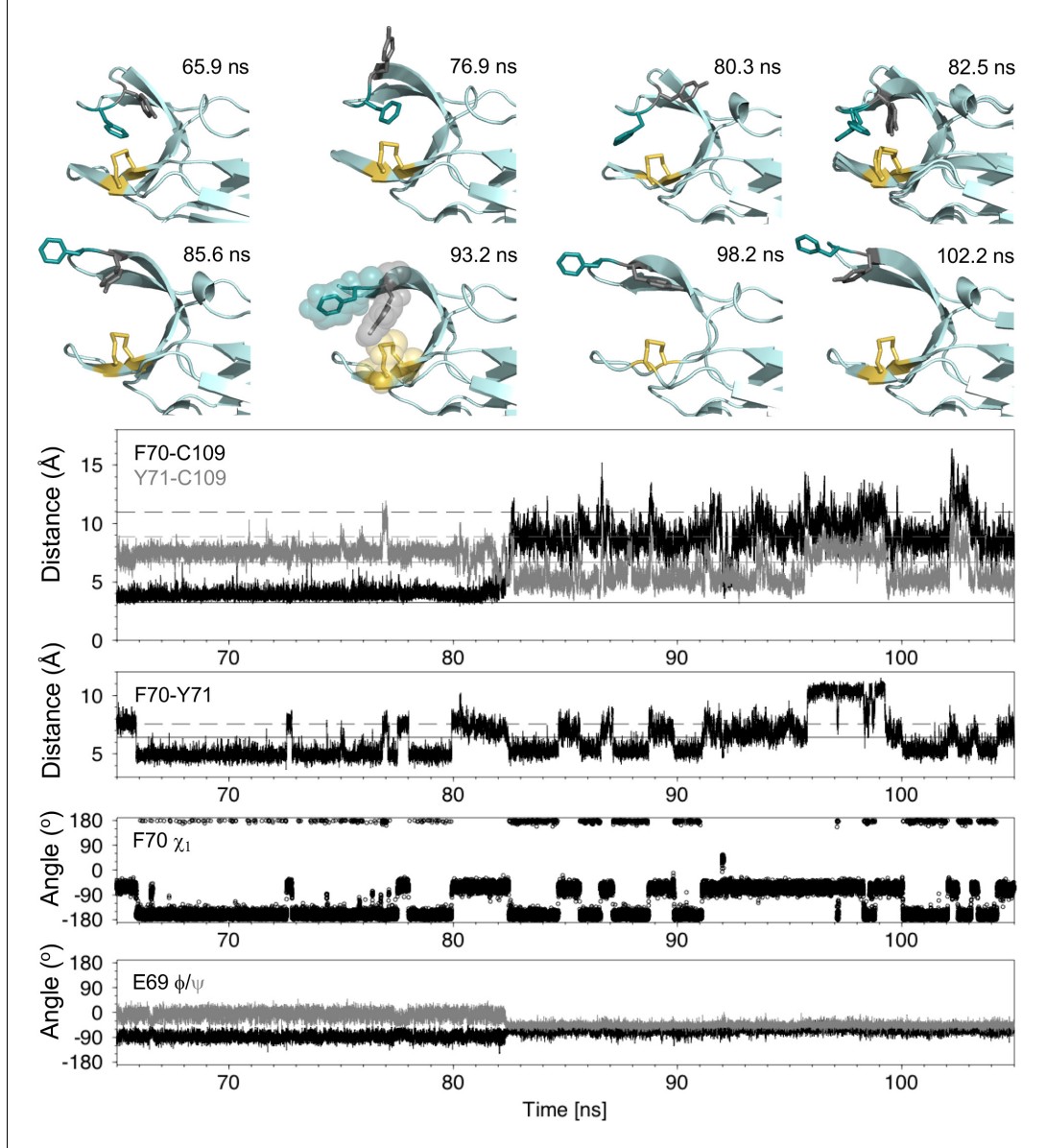

**Figure 5.** Analysis of the major cap-loop opening event observed in MD simulations of nDsbD$_{ox}$. Top: Snapshots of structures generated from the MD simulation between 65 and 105 ns show the range of cap loop conformations and, in particular, the F70 and Y71 side chain positions that are sampled. The backbone of nDsbD is shown as a cartoon in pale cyan. F70, Y71 and C103/C109 are shown in a stick representation and colored deep teal, grey and yellow, respectively. Two structures are overlaid in the 82.5 ns panel; these illustrate the conversion of F70 from the gauche- to the trans conformation observed simultaneously with loop opening. A surface representation is used in the 93.2 ns snapshot to illustrate how Y71 blocks closing of the cap loop. Bottom: Structural parameters are plotted as a function of simulation time. The distance from the center of the rings of F70 and Y71 to C109 Sγ are shown in black and grey in the first panel. The distance between the ring centres of F70 and Y71 are shown in the second panel. The solid and dashed lines show these distances in the closed (1L6P) and open (1VRS) X-ray structures, respectively. In the third and fourth panels, the F70 $\chi_1$ torsion angle and the E69 $\phi$ and $\psi$ torsion angles are shown. These and additional structural parameters are shown in the figure supplements for the entire loop opening period (60–200 ns) and for a second 22 ns loop opening event.

The online version of this article includes the following figure supplement(s) for figure 5:

**Figure supplement 1.** Analysis of distances and torsion angles for 30 ns of a molecular dynamics trajectory showing a 22 ns cap loop opening.

**Figure supplement 2.** Analysis of distances for 140 ns of a molecular dynamics trajectory showing cap loop opening.

**Figure supplement 3.** Analysis of $\chi_1$ and $\chi_2$ torsion angles for 140 ns of a molecular dynamics trajectory showing cap loop opening.

**Figure supplement 4.** Analysis of $\phi$ and $\psi$ torsion angles for 140 ns of a molecular dynamics trajectory showing cap loop opening.

**Figure supplement 5.** Snapshots of structures generated from the molecular dynamics simulations of nDsbD$_{ox}$.

**Figure supplement 6.** Sequence alignment of the cap-loop and active-site regions of nDsbD from β- and γ-proteobacteria.

the backbone hydrogen bonds which define the antiparallel β-hairpin of the cap loop and which involve D68-G72 and H66-S74, are always retained in the open loop structures.

We have also investigated the dynamics of Y71 in the simulations of loop closing, and in particular those simulations in which the loop fails to close completely within 10 ns. During these trajectories Y71 adopts conformations in which it sits between F70 and C109 thus blocking complete closure of the loop as observed, for example, at 93.2 ns in the major opening event and at 2.8 ns in the incomplete loop closing seen in *Figure 3G* (*Figure 5* and *Figure 5—figure supplement 5*).

In the light of the important roles identified for both F70 and Y71 in cap loop opening, we investigated the conservation of cap loop residues using bioinformatics. We have used 134 representative sequences, one from each bacterial genus, for our sequence alignment (*Figure 5—figure supplement 6*). We find that the cap-loop motif is more conserved than the flanking β-strands, in agreement with the central role that this structural motif plays in nDsbD function. In 53% of the sequences an F or a Y is found at position 70; in a further 36.4% of the sequences a residue with a relatively bulky side chain (K/I/L/E/V/T/N), which would shield the cysteines from solvent, is found at position 70. Interestingly, in 85% of sequences an F or a Y is found at position 71; in a further 5.2% of the sequences position 71 is occupied by a bulky hydrophobic residue (I or L). This high conservation of an aromatic amino acid observed for this position further highlights the key role played by Y71 in contributing to loop opening and, more importantly, in preventing rapid loop closure, thus likely facilitating the interaction of nDsbD with cDsbD prior to reductant transfer.

## NMR relaxation dispersion experiments

Using $^{15}N$ NMR relaxation dispersion experiments (*Baldwin and Kay, 2009*; *Loria et al., 1999*) we also detected a difference in the μs-ms timescale dynamics of reduced and oxidized nDsbD. For nDsbD$_{red}$, no evidence for μs-ms dynamics in the cap loop or the active site were found (*Figure 6A–C*). This observation is consistent with the absence of a significantly populated open state of nDsbD$_{red}$. By contrast, in nDsbD$_{ox}$, eight residues, V64, W65, E69, F70, Y71, G72, K73 and S74, all in the cap-loop region, showed strong relaxation dispersion effects (*Figures 6D–F* and *7A*). These results are consistent with previous observations about peak intensities in $^{1}H$-$^{15}N$ HSQC spectra (*Mavridou et al., 2012*). The peaks of C103, C109 and Y110 are not visible and the peak of A104 is very weak in the spectrum of nDsbD$_{ox}$, but not nDsbD$_{red}$ (*Mavridou et al., 2012*), likely due to extensive chemical exchange processes in nDsbD$_{ox}$ leading to $^{1}H^{N}$ and/or $^{15}N$ peak broadening. Thus, the cap loop of nDsbD$_{ox}$, and likely also the active-site cysteines, exchange between different conformations on the μs-ms timescale, whereas nDsbD$_{red}$ adopts a single conformation on this timescale.

Analysis of the relaxation dispersion data collected at 25°C (using CATIA [*Vallurupalli et al., 2008*; *Baldwin et al., 2011a*; *Alderson et al., 2019*]) showed that nDsbD$_{ox}$ undergoes a single global exchange process, between a dominant major state (p$_A$98%) and an alternative minor state (p$_B$2%) (*Figure 6—figure supplement 1—source data 1*). To corroborate that a single exchange process gives rise to the relaxation dispersion curves, we repeated the experiments at four additional temperatures (10, 15, 20°C and 35°C). If the residues underwent different exchange processes we would expect them to show a differential response to temperature (*Grey et al., 2003*). The relaxation dispersion data for all residues in the cap loop recorded at the five temperatures could be fitted simultaneously (*Baldwin et al., 2011a*; *Alderson et al., 2019*; *Figure 6—figure supplement 1A* and *Figure 6—figure supplement 1—source data 1*). This confirmed that a single process underlies the μs-ms chemical exchange in the cap loop of nDsbD$_{ox}$. In addition, the analysis of the data at all five temperatures yielded a detailed description of the thermodynamics and kinetics of the exchange process (*Figure 6—figure supplement 1B* and *Figure 6—figure supplement 1—source data 1*). The minor 'excited' state is enthalpically, but not entropically, more favorable than the major ground state of nDsbD$_{ox}$. The fits also demonstrated that no enthalpic barrier separates the ground and excited state of nDsbD$_{ox}$. This absence of an enthalpic barrier between the two states might mean that no favorable interactions have to be broken in the transition state, but that the diffusion process itself may limit the speed of the transition (*Dill and Bromberg, 2010*).

Analysis of the chemical shift differences determined from the relaxation dispersion experiments suggests that the conformation adopted by the cap loop in the minor state of nDsbD$_{ox}$ might be similar to the conformation of nDsbD$_{red}$. To interpret the chemical shift differences, we calculated the shift changes expected if the minor state features a disordered cap loop (*Nielsen and Mulder,*

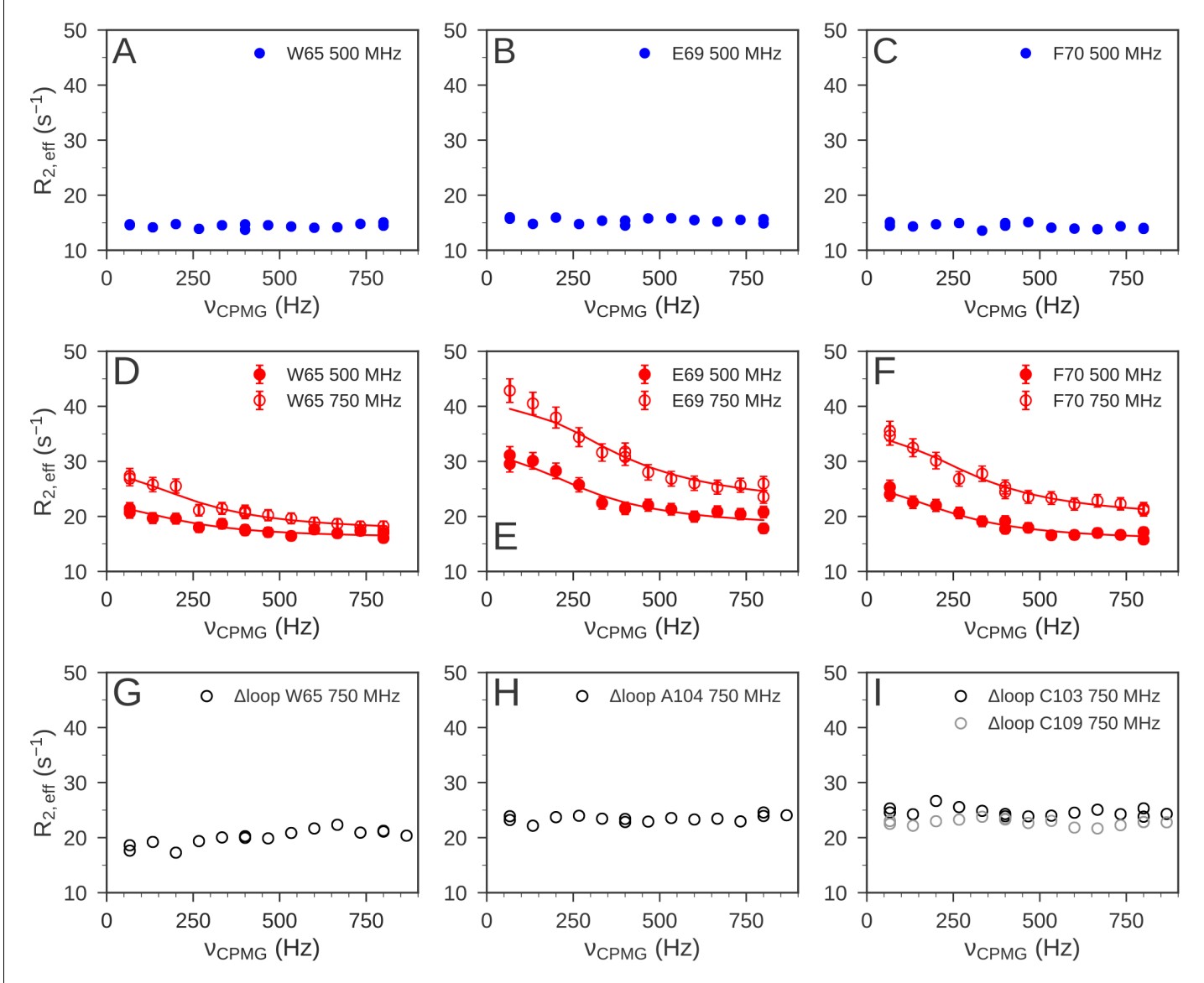

**Figure 6.** $^{15}$N relaxation dispersion experiments were used to determine the µs-ms dynamics of nDsbD. For nDsbD$_{red}$, flat relaxation dispersion profiles were obtained as illustrated for (**A**) W65, (**B**) E69 and (**C**) F70 in the cap loop. For nDsbD$_{ox}$ clear relaxation dispersion effects were measured for (**D**) W65, (**E**) E69 and (**F**) F70; error bars were determined as described in *Figure 6—figure supplement 1—source data 1*. In the oxidized cap-loop deletion mutant (Δloop-nDsbD$_{ox}$), flat relaxation dispersion profiles were obtained for (**G**) W65, (**H**) A104 and (**I**) C103/C109 in the active site. Experimental data are represented by circles and fits to the data for wild-type nDsbD$_{ox}$ are shown by solid lines.

The online version of this article includes the following source data and figure supplement(s) for figure 6:

**Figure supplement 1.** Analysis of the relaxation dispersion experiments at multiple temperatures.

**Figure supplement 1—source data 1.** Parameters from a multi-temperature analysis of the relaxation dispersion data.

**Figure supplement 2.** Comparison of the X-ray structures of wild-type nDsbD$_{ox}$ (**A**) and Δloop-nDsbD$_{ox}$ (**B**).

**Figure supplement 2—source data 1.** Crystallographic data collection and refinement statistics for Δloop-nDsbD$_{ox}$.

*2018*). Such a state would include more open conformations and such structures might facilitate the binding of the nDsbD interaction partners. The lack of correlation in *Figure 7B* demonstrates that the cap loop does not become disordered in the less populated minor state. The $^{15}$N chemical shift differences between the major and minor states do, however, resemble the chemical shift differences between the oxidized and reduced isoforms of nDsbD (*Figure 7C*; *Mavridou et al., 2012*). Nonetheless, the correlation in *Figure 7C* is not perfect; the differences between the chemical shifts

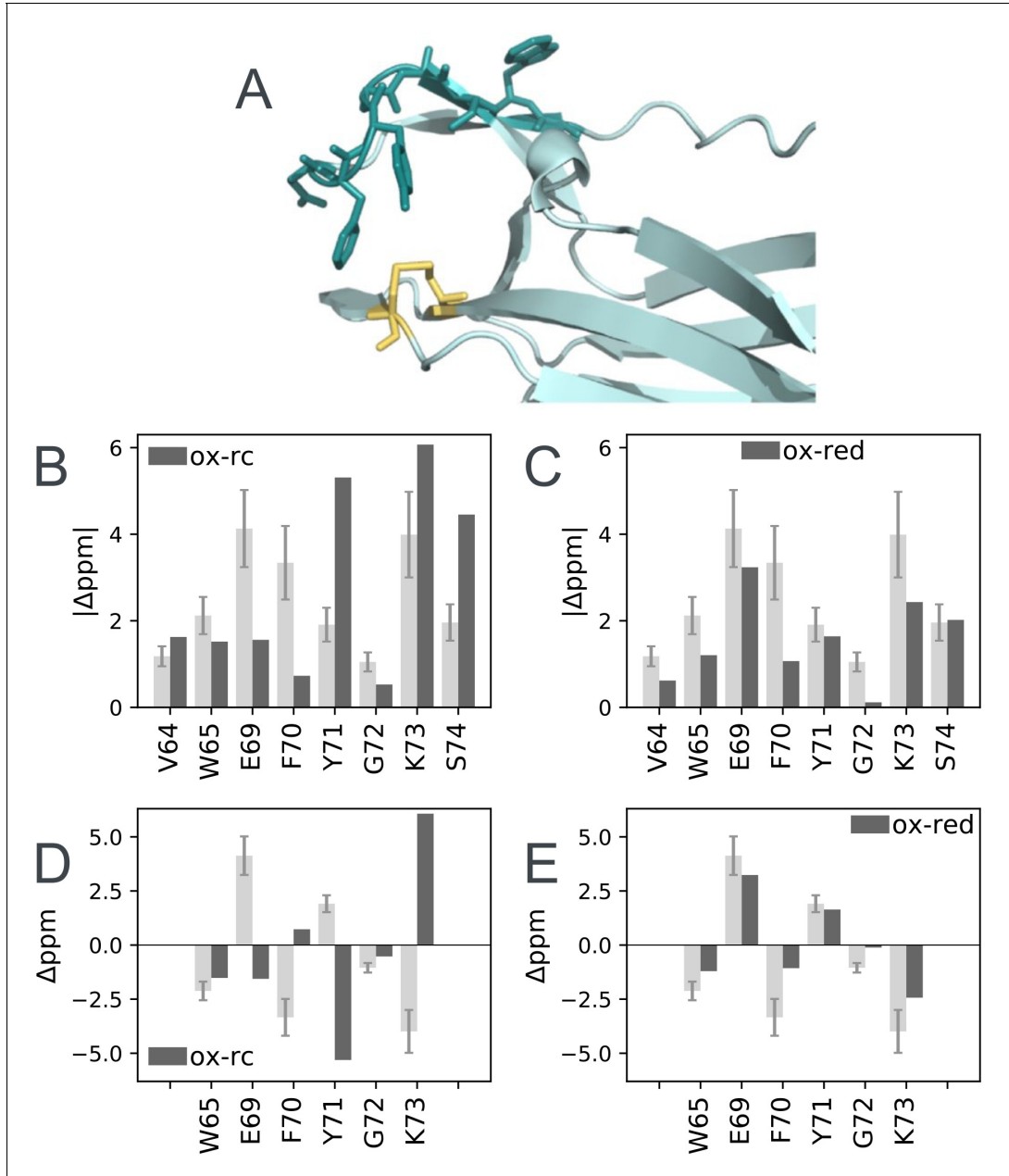

**Figure 7.** Structural characterization of μs-ms dynamics in nDsbD$_{ox}$. (A) Residues 64, 65 and 69–74 with chemical shift differences between the major and minor states of >1 ppm are shown in a stick representation in dark teal. C103 and C109 are shown in yellow. The magnitude of the $^{15}$N chemical shift differences between the major and minor states are compared in (B) with the magnitude of predicted shift differences between the native and random coil states and in (C) with the magnitude of the experimental shift difference between nDsbD$_{ox}$ and nDsbD$_{red}$. Shift differences between the major and minor states for W65, E69, F70, Y71, G72 and K73, for which the sign has been obtained experimentally, are compared with the predicted shift differences between the native and random coil states in (D) and between nDsbD$_{ox}$ and nDsbD$_{red}$ in (E). The experimentally determined chemical shift differences are shown in light grey, with error bars. Random coil shifts were predicted using POTENCI (*Nielsen and Mulder, 2018*). Details of the fitting procedures used for the dispersion data and the determination of errors in the chemical shift differences are presented in *Figure 6—figure supplement 1* and *Figure 6—figure supplement 1—source data 1* and *Figure 7—source data 1*.

The online version of this article includes the following source data for figure 7:

**Source data 1.** Determination of the sign of the $^{15}$N chemical shift difference between the excited state and the ground state determined by comparison of HSQC and HMQC spectra collected at 750 MHz.*

of the minor state of nDsbD$_{ox}$ and those of nDsbD$_{red}$ likely reflect the local electronic and steric differences between having a disulfide bond (in the minor state of nDsbD$_{ox}$) and two cysteine thiols (in nDsbD$_{red}$) in the active site. Importantly, the sign of the chemical shift differences between the major and minor state, which could be determined experimentally for W65, E69, F70, Y71, G72 and K73, further indicated that the minor state structure of the cap loop of nDsbD$_{ox}$ is not disordered (*Figure 7D* and *Figure 7—source data 1*), but instead may be similar to that adopted in nDsbD$_{red}$ (*Figure 7E* and *Figure 7—source data 1*).

## Coupling between the active site and the cap loop gives rise to the alternative minor state of oxidized nDsbD

The alternative minor state of nDsbD$_{ox}$, which we detected by NMR relaxation dispersion, can also be attributed to coupling between the dynamics of the active-site cysteine residues and the cap loop. In this alternative minor state, the C103 and C109 side chains likely adopt a 'reduced-like' conformation. Both side chains adopt a gauche- conformation ($\chi_1 \sim -60°$) in nDsbD$_{ox}$, but a trans conformation ($\chi_1 \sim 180°$) in nDsbD$_{red}$. In MD simulations of both redox states, rare excursions of one or the other $\chi_1$, but never both, to the alternative conformation are observed (*Figure 4C/F*). We postulate that in the minor state of nDsbD$_{ox}$, both of the cysteine side chains adopt the 'reduced-like' trans conformation. In this state, the cap loop could pack tightly onto the active-site disulfide bond, as observed in MD simulations of nDsbD$_{red}$. The backbone $^{15}$N chemical shifts of this minor state consequently resemble those of the reduced protein. Model building of a disulfide bond into the reduced 3PFU crystal structure followed by energy minimization with XPLOR (*Brunger, 1992*) showed that the resulting 'oxidized' structure maintains the 'reduced-like' trans conformation ($\chi_1 \sim -140°$) for the cysteine side chains and the gauche- conformation for F70 (*Figure 8B*). This 'reduced-like' conformation for nDsbD$_{ox}$, with a more tightly packed active site and a cap loop which lacks frustration, might be enthalpically more favorable and entropically less favorable than the major state of nDsbD$_{ox}$ as shown by the analysis of relaxation dispersion data collected at multiple temperatures (*Figure 6—figure supplement 1* and *Figure 6—figure supplement 1—source data 1*). Previously, NMR relaxation-dispersion experiments and MD simulations have detected complex dynamics in BPTI stemming from one of its disulfide bonds (*Grey et al., 2003*; *Shaw et al., 2010*; *Xue et al., 2012*), further suggesting that the disulfide bond shapes the μs-ms dynamics of the active-site and cap-loop regions of nDsbD$_{ox}$.

## NMR relaxation dispersion experiments using a loop deletion variant

We confirmed the tight coupling between the dynamics of the cap loop and the active site by NMR relaxation dispersion experiments using an nDsbD variant. We generated an nDsbD protein with a truncated cap loop in which the eight residues H66-K73 were replaced by A-G-G (Δloop-nDsbD). The X-ray structure of oxidized Δloop-nDsbD shows that deletion of the cap loop does not cause any significant changes in structure; a Cα RMSD of 0.25 Å is obtained for residues 12–122 (*Figure 6—figure supplement 2* and *Figure 6—figure supplement 2—source data 1*). Nevertheless, in the absence of F70, the shorter cap loop can no longer closely pack onto the disulfide bond. As a result, no relaxation dispersion is detected in the truncated cap loop variant, as shown for W65 (*Figure 6G*). For A104 (A99), a much sharper amide peak is observed in the HSQC spectrum of Δloop-nDsbD$_{ox}$ and this residue shows a flat dispersion profile (*Figure 6H*). Importantly, the $^1$H$^N$-$^{15}$N peaks for C103 (C98) and C109 (C104), which are not observed in the wild-type oxidized protein, can be detected for Δloop-nDsbD$_{ox}$. Finally, $^{15}$N relaxation dispersion profiles for these cysteines are also flat (*Figure 6I*). Thus, truncation of the cap loop alters the μs-ms dynamics of the cap-loop region and the active-site cysteines, further supporting our proposed model of tight coupling between the cap loop and active-site cysteines.

## Conclusion

We have studied oxidized and reduced nDsbD using NMR experiments and microsecond duration MD simulations, and have demonstrated that the conformational dynamics of the active-site cysteines and the cap loop are coupled and behave in an oxidation-state-dependent manner. We showed that the cap loop is rigid in nDsbD$_{red}$, whilst it exhibits more complex dynamics in nDsbD$_{ox}$ (*Figure 8*). Although the cap loop is predominantly closed and ordered in both oxidation states, for

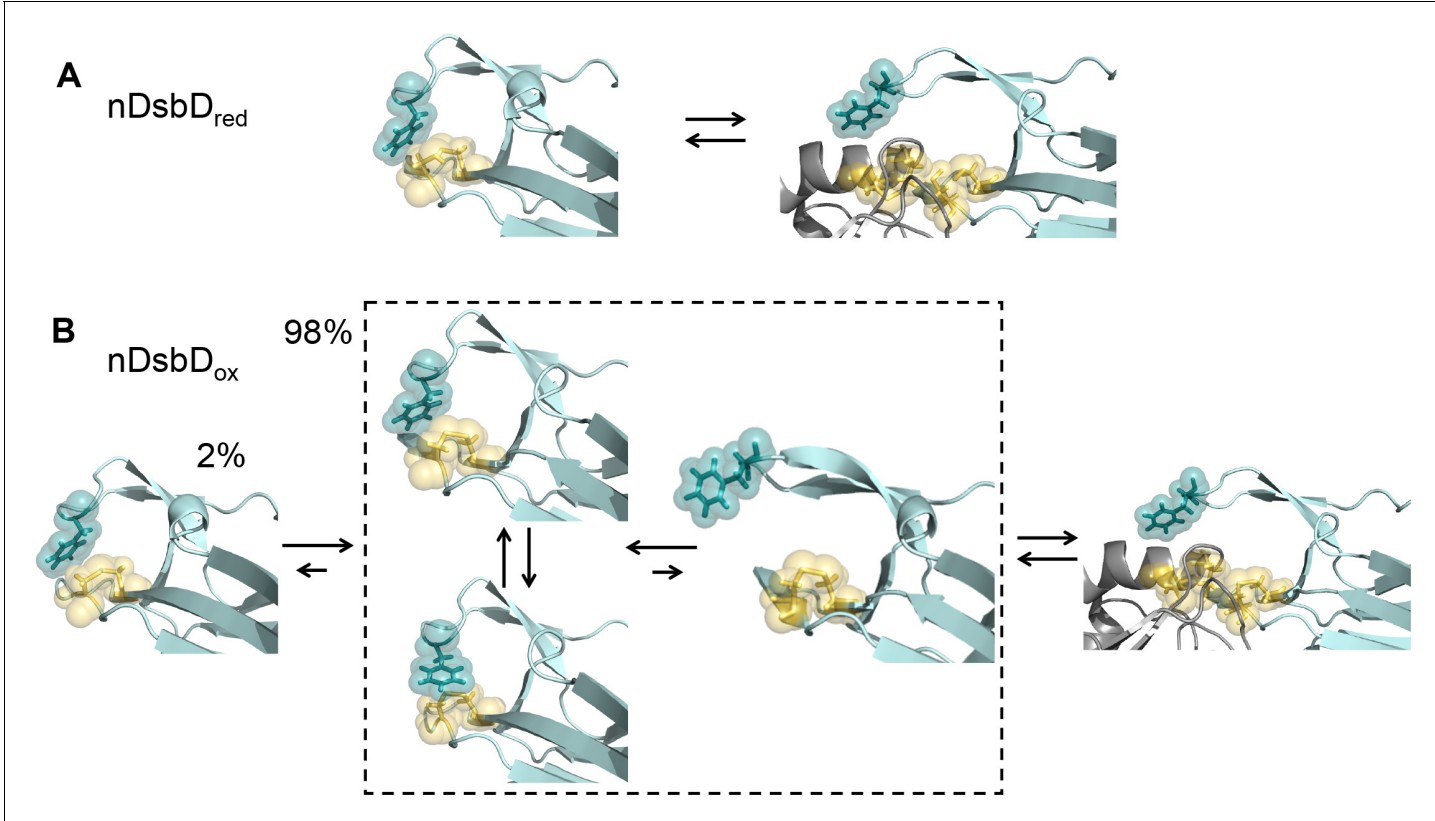

**Figure 8.** Local frustration occurs in oxidized, but not in reduced, nDsbD. (**A**) The cap loop in nDsbD$_{red}$ maintains a closed conformation shielding the active-site cysteine thiols. The cap loop may only open to expose the cysteine thiols via an induced-fit mechanism when it encounters its legitimate binding partners, CcmG, DsbC and DsbG. (**B**) nDsbD$_{ox}$ undergoes more complex dynamics as a result of local structural frustration. F70 interacts with C109 Sγ both in its gauche- (top left structure in dashed box) and trans (lower left structure in dashed box) orientations. These mutually competing interactions destabilize the packing of the cap loop onto the active site. Consequently, open conformations (right-hand structure in dashed box) are accessible in nDsbD$_{ox}$ and these would allow nDsbD$_{ox}$ to form complexes with its legitimate binding partner, cDsbD$_{red}$, via a conformational selection mechanism. The left-hand structure shows a reduced-like minor (2%) conformation identified by relaxation dispersion experiments and populated on a slower timescale. The backbone of nDsbD and cDsbD are shown in cyan and grey, respectively. F70 (dark teal), C103 and C109 (yellow) are shown with sticks and a surface representation. (**A**) is from the 3PFU X-ray structure of nDsbD$_{red}$, the left-hand panel in (**B**) is the energy minimized 3PFU structure with a disulfide bond incorporated, the structures in the dashed rectangle in (**B**) are 'snapshots' from the MD simulations of nDsbD$_{ox}$. The bound complexes shown on the right in (**A**) and (**B**) are based on the 1VRS X-ray structure.

nDsbD$_{red}$ we found no evidence for a significant population with an open cap loop. Our results argue that the cap loop of nDsbD$_{red}$ remains closed at all times, to protect the active-site cysteine thiol groups from non-cognate reactions in the oxidizing environment of the periplasm. As such, the nDsbD$_{red}$ cap loop may only open to expose the cysteine thiols via an induced-fit mechanism when it encounters its legitimate binding partners, for example CcmG, DsbC and DsbG. In nDsbD$_{ox}$, on the other hand, the cap loop opens spontaneously in MD simulations and while fully open conformations of the cap loop tended to close, partially open structures were stable. Thus, nDsbD$_{ox}$ may form a complex with its sole legitimate binding partner, cDsbD$_{red}$, via a conformational selection mechanism. Notably, NMR relaxation dispersion experiments provided no evidence for a long-lived open or disordered conformation of the cap loop as might have been expected. Instead, a minor state with a 'reduced-like' conformation in the active site and cap loop is observed solely for nDsbD$_{ox}$.

Importantly, combining NMR experiments and MD simulations, we showed how local frustration (*Ferreiro et al., 2007*; *Ferreiro et al., 2011*) in nDsbD$_{ox}$, but not in nDsbD$_{red}$, gives rise to differences in the dynamics of the two redox isoforms, on multiple timescales, providing insight into how this could affect protein function. Local frustration has been highlighted in the interaction interfaces of many protein complexes (*Ferreiro et al., 2007*), where mutually exclusive favorable interactions result in a dynamic equilibrium of competing structures and determine which parts of a protein are

flexible. In nDsbD$_{red}$, the cap loop packs perfectly onto the active-site cysteines and the closed conformation is stable, something that is crucial in the very oxidizing environment in which the domain has to provide electrons. By contrast, in nDsbD$_{ox}$, the disulfide bond disrupts this tight fit of the aromatic ring of F70. Consequently F70 is no longer locked in a single conformation and switches freely between gauche- and trans orientations. This results in increased flexibility for both F70 and Y71 and drives occasional opening of the cap loop, which in this oxidation state might facilitate nDsbD$_{ox}$ receiving electrons from its partner. This is further enabled by a possible role of Y71 in preventing rapid loop closing. The observation from sequence alignments that the general architecture of the cap-loop region in β- and γ-proteobacteria is conserved, highlights that our observations in *E. coli* nDsbD can be generalized for many other bacterial species where reductant provision is driven by DsbD in the cell envelope.

Overall, our results show that it is the oxidation state of the cysteine pair in nDsbD that acts as a 'switch' controlling the presence or absence of local frustration in its active site. This oxidation-state-dependent 'switch' determines the dynamic behavior of the cap loop which translates into optimized protein-protein interaction and biological activity. We propose that redox-state modulation of local frustration, such as that observed here for nDsbD, may play a wider role in biomolecular recognition in other pathways involving redox proteins which are ubiquitous in all organisms. Ultimately, the understanding gained from studies like this would allow controlled introduction of frustration, mimicking intricate natural systems such as DsbD, for more translational applications such as the production of synthetic molecular devices.

# Materials and methods

## Key resources table

| Reagent type (species) or resource | Designation | Source or reference | Identifiers | Additional information |
|---|---|---|---|---|
| Gene (*Escherichia coli*) | DSBD | | DSBD_ECOLI | UniprotID P36655 |
| Strain, strain background (*Escherichia coli*) | BL21(DE3) | Stratagene | | Competent cells |
| Recombinant DNA reagent | pDzn3 | DOIs:10.1007/s12104-011-9347-9 and 10.1074/jbc.M805963200 | N-terminal domain of DsbD (nDsbD) (L2-V132) | Plasmid for wild-type nDsbD expression |
| Recombinant DNA reagent | pDzn8 | This study. | nDsbD (L2-V132) with residues H66-K73 replaced by AGG | Plasmid for Δ-loop nDsbD expression |
| Sequence-based reagent | Δ-loop nDsbD | This study. Synthesized by Sigma Aldrich. | | PCR primer (forward): 5'-(AGGAAGCGAGATTTAC CGCGATCGGCTG)−3' |
| Sequence-based reagent | Δ-loop nDsbD | This study. Synthesized by Sigma Aldrich. | | PCR primer (reverse): 5'-(CCGGCCCAGACGCCTTGC GGCAGCTGC)−3' |
| Peptide, recombinant protein | Wild-type nDsbd | DOIs:10.1007/s12104-011-9347-9 and 10.1074/jbc.M805963200 | N-terminal domain of DsbD (nDsbD) (L2-V132) | Purified using a thrombin-cleavable C-terminal poly-His tag |
| Peptide, recombinant protein | Δ-loop nDsbd | This study. | nDsbD (L2-V132) with residues H66-K73 replaced by AGG | Purified using a thrombin-cleavable C-terminal poly-His tag |
| Commercial assay or kit | Sigma Thrombin CleanCleave Kit | Sigma-Aldrich | | Used for removal of the poly-His tag |
| Chemical compound, drug | KOD Hot Start DNA polymerase | Novagen | | Used for PCR |

*Continued on next page*

*Continued*

| Reagent type (species) or resource | Designation | Source or reference | Identifiers | Additional information |
|---|---|---|---|---|
| Chemical compound, drug | Pf1 phage | ASLA Biotech AB | | Used for partial alignment for RDC measurements |
| Software, algorithm | NMRPipe | DOI: 10.1007/BF00197809 | | Processing of NMR data |
| Software, algorithm | CCPN Analysis (version 2.4.2) | DOI: 10.1002/prot.20449 | RRID:SCR_016984 | Analysis of NMR spectra |
| Software, algorithm | CATIA | DOIs:10.1073/pnas.0804221105 and 10.1016/j.jmb.2011.07.017 and 10.1038/s41467-019-08557-8 | https://www.ucl.ac.uk/hansen-lab/catia/ | Global fitting of relaxation dispersion NMR data |
| Software, algorithm | GROMACS 4.5 | DOI: 10.1002/jcc.20291 | RRID:SCR_014565 | Used for MD simulations |
| Software, algorithm | WHAT-IF | DOI: 10.1093/nar/gkq453 | https://swift.cmbi.umcn.nl/whatif | Used to add missing atoms prior to MD simulations |
| Software, algorithm | Visual Molecular Dynamics (VMD) | DOI: 10.1016/0263-7855(96)00018-5 | RRID:SCR_001820 | Used for analysis of distances, angles, hydrogen bonds in MD simulations and for rendering protein structures |
| Software, algorithm | MDAnalysis Python Library | DOI: 10.1002/jcc.21787 | | Used for analysis of distances, angles, hydrogen bonds in MD simulations |
| Software, algorithm | blast 2.2.28+ (blastp) | NCBI | RRID:SCR_001010 https://blast.ncbi.nlm.nih.gov/Blast.cgi | Used to search for DsbD protein homologs |
| Software, algorithm | fasttree 2.1.7 | DOI: 10.1371/journal.pone.0009490 | RRID:SCR_015501 | Used for building a phylogenetic tree |
| Software, algorithm | muscle 3.8.31 | DOI: 10.1093/nar/gkh340 | RRID:SCR_011812 | Used for sequence alignment |
| Software, algorithm | Pymol Version 2.3.5 | Schrodinger, LLC | RRID:SCR_000305 | Used for rendering protein structural figures |
| Software, algorithm | Denzo/ Scalepack | DOI: 10.1016/S0076-6879(97)76066-X | | Used for processing and scaling of X-ray data |
| Software, algorithm | MOLREP | DOI: 10.1107/S0907444996012255 | | Used for molecular replacement in solving X-ray structure |
| Software, algorithm | Phenix.refine | DOI: 10.1107/S0907444909052925 | RRID:SCR_016736 | Used for refinement of X-ray structure |
| Software, algorithm | Coot | DOI: 10.1107/S0907444910007493 | RRID:SCR_014222 | Used for manual fitting of electron density in X-ray structure |

## Construction of plasmids and protein production

The plasmid pDzn3, described in previous work (*Mavridou et al., 2012*; *Mavridou et al., 2009*), encodes isolated wild-type nDsbD (L2-V132) bearing a thrombin-cleavable C-terminal poly-histidine tag. This construct was used as a template to produce a cap-loop deletion variant of nDsbD (Δloop-nDsbD), where residues H66-K73 were replaced by the amino acid sequence A-G-G. Site-directed mutagenesis (ExSite, Stratagene) was performed using oligonucleotides 5'-(AGGAAGCGAGATTTACCGCGATCGGCTG)–3' and 5'-(CCGGCCCAGACGCCTTGCGGCAGCTGC)–3'. PCR was performed with KOD Hot Start DNA polymerase (Novagen) and oligonucleotides were synthesized by Sigma Aldrich. DNA manipulations were conducted using standard methods. The resulting plasmid was named pDzn8. nDsbD and Δloop-nDsbD were expressed using BL21(DE3) cells (Stratagene) and purified from periplasmic extracts of *E. coli* using a C-terminal poly-histidine tag, as described previously (*Mavridou et al., 2012*; *Mavridou et al., 2007*). Thrombin cleavage of the affinity tag was

performed using the Sigma Thrombin CleanCleave Kit (Sigma) according to the manufacturer's instructions.

## NMR spectroscopy

NMR experiments were conducted using either home-built 500, 600 and 750 MHz spectrometers equipped with Oxford Instruments Company magnets, home-built triple-resonance pulsed-field gradient probeheads and GE/Omega data acquisition computers and software, or using a 750 MHz spectrometer equipped with a Bruker Avance III HD console and a 5 mm TCI cryoprobe. All experiments were conducted at 25°C and at pH 6.5 in 95% $H_2O$/5% $D_2O$, unless stated otherwise. Spectra were processed using NMRPipe (*Delaglio et al., 1995*) and analyzed using CCPN Analysis (*Skinner et al., 2016*; *Vranken et al., 2005*).

Residual dipolar couplings (RDCs) were measured using Pf1 phage (ASLA Biotech AB) on separate samples for oxidized and reduced nDsbD. RDCs were measured at 600 MHz for 0.5 mM nDsbD with 10 mg/ml Pf1 phage, 2 mM $K_2PO_4$, 0.4 mM $MgCl_2$, 0.01% $NaN_3$ and 10% $D_2O$. Measurements were also carried out for isotropic solutions prior to the addition of the Pf1 phage. RDCs were measured using the InPhase-AntiPhase (IPAP) approach (*Ottiger et al., 1998*). The $F_2$ $^1H$ and $F_1$ $^{15}N$ dimensions were recorded with 128 scans per increment, sweep widths of 7518.8 and 2000.0 Hz, respectively, and 1024 and 128 complex points, respectively. The quality factor (Q) that describes the agreement between calculated and observed RDCs was determined using the procedure of *Ottiger et al., 1997*.

{$^1H$}-$^{15}N$ heteronuclear NOE, $^{15}N$ $T_1$ and $T_2$ data (*Boyd et al., 1990*; *Kay et al., 1989*; *Palmer et al., 1992*) were measured at 600 MHz. The {$^1H$}-$^{15}N$ heteronuclear NOE was also measured at 500 MHz. For the $T_1$ measurement, spectra with 14 different relaxation delays, ranging from 20 ms to 2 s, were collected. The $T_2$ was measured by recording spectra with 14 relaxation delays between 8 ms and 400 ms, with a Carr-Purcell Meiboom Gill (CPMG) delay $\tau_{CPMG}$ of 0.5 ms. For the $T_1$ and $T_2$ measurements, a recycle delay of 2 s was used. The {$^1H$}-$^{15}N$ NOE was measured by comparing peak heights in interleaved spectra recorded with and without saturation of the protons for 3 and 4 s at 500 and 600 MHz, respectively. The $F_2$ $^1H$ and $F_1$ $^{15}N$ dimensions were recorded with sweep widths of 7518.8 and 1785.7 Hz, respectively, and with 1024 and 128 complex points, respectively. The $T_1$ and $T_2$ and NOE experiments were collected with 16, 16 and 128 scans per increment, respectively.

$^{15}N$ relaxation data were collected for 107 and 117 of the 126 non-proline residues of oxidized and reduced nDsbD, respectively. Data for the remaining residues could not be measured due to weak or overlapping peaks. $T_1$, $T_2$ and {$^1H$}-$^{15}N$ NOE values were determined as described previously (*Smith et al., 2013*). Uncertainties in the $T_1$, $T_2$ and {$^1H$}-$^{15}N$ NOE values were estimated from 500 Monte Carlo simulations using the baseline noise as a measure of the error in the peak heights (*Smith et al., 2013*). Relaxation data were analyzed using in-house software (*Smith et al., 2013*); this incorporates the model-free formalism of Lipari and Szabo (*Lipari and Szabo, 1982a*; *Lipari and Szabo, 1982b*) using spectral density functions appropriate for axially symmetric rotational diffusion (*Abragam, 1961*) and non-colinearity of the N-H bond vector and the principal component of the $^{15}N$ chemical shift tensor (*Boyd and Redfield, 1998*) with model selection and Monte Carlo error estimation as described by *Mandel et al., 1995*. Calculations were carried out using an N-H bond length of 1.02 Å, a $^{15}N$ chemical shift anisotropy, ($\sigma_{\parallel}$ - $\sigma_{\perp}$), of −160 ppm, and a $D_{\parallel}/D_{\perp}$ ratio of 2.0.

$^{15}N$ relaxation-dispersion experiments (*Loria et al., 1999*) were recorded at multiple magnetic field strengths and temperatures. Recycle delays of 1.2 and 1.5 s were used at 500 and 750 MHz, respectively. Typically 12 to 14 spectra were recorded with refocusing fields, $\nu_{CPMG}$, between 50 and 850 Hz and two reference spectra (*Mulder et al., 2001*) were collected to convert measured peak intensities into relaxation rates. At 15°C and 25°C, experiments were collected at both 500 and 750 MHz. Experiments at 10°C, 20°C and 35°C were recorded at 500 MHz only. Δloop-nDsbD, was studied at 25°C at 750 MHz. The experiments were analyzed by global fitting using CATIA (Cpmg, Antitrosy, and Trosy Intelligent Analysis, https://www.ucl.ac.uk/hansen-lab/catia/) (*Vallurupalli et al., 2008*; *Baldwin et al., 2011a*; *Alderson et al., 2019*) using transition state theory to restrict the rates and linear temperature-dependent changes in chemical shifts when analyzing data across multiple temperatures (*Vallurupalli et al., 2008*; *Baldwin et al., 2011a*; *Alderson et al., 2019*).

The HSQC and HMQC pulse sequences developed by *Skrynnikov et al., 2002* were used to determine the sign of the chemical shift differences between the major and minor states. Bruker

versions of the sequences were kindly provided by Prof. LE Kay. Experiments were recorded in triplicate at 750 MHz using a spectrometer equipped with a cryoprobe. 48 scans were collected per $^{15}$N increment, using $^1$H and $^{15}$N sweep widths of 11904.762 and 2493.767 Hz, respectively, and with 1024 and 175 complex points in the $^1$H and $^{15}$N dimensions, respectively.

## Molecular dynamics simulations

MD simulations were run using GROMACS 4.5 (*Van Der Spoel et al., 2005*). Simulations were started from the closed crystal structures of reduced nDsbD (nDsbD$_{red}$) (PDB: 3PFU) (*Mavridou et al., 2011*) and oxidized nDsbD (nDsbD$_{ox}$) (PDB 1L6P) (*Goulding et al., 2002*). For each redox state of nDsbD, four simulations of 200 ns and ten simulations of 20 ns, all with different initial velocities, were run giving a combined duration of 1 μs for each protein. Trajectories were also initiated from open structures (PDB: 1VRS chain B) (*Rozhkova et al., 2004*); eight and ten 10 ns trajectories were run for nDsbD$_{red}$ and nDsbD$_{ox}$, respectively. Missing side-chain atoms were added using the WHAT-IF Server (https://swift.cmbi.umcn.nl/whatif) (*Hekkelman et al., 2010*). The histidine side chains were protonated and the cysteine side chains (C103 and C109) in the active site of nDsbD$_{red}$ were represented as thiols; these choices were based on pH titrations monitored by NMR (*Mavridou et al., 2012*). The protein was embedded in rhombic dodecahedral boxes with a minimum distance to the box edges of 12 Å at NaCl concentrations of 0.1 M. Trajectories using a larger distance of 15 Å to the box edges showed no significant differences. The CHARMM 22 force field (*MacKerell et al., 1998*) with the CMAP correction (*Mackerell et al., 2004*) and the CHARMM TIP3P water model was used. Electrostatic interactions were calculated with the Particle Mesh Ewald method (PME) (*Essmann et al., 1995*). The Lennard-Jones potential was switched to zero between 10 Å and 12 Å (*Bjelkmar et al., 2010*). The length of bonds involving hydrogen atoms was constrained using the PLINCs algorithm (*Hess, 2008*). The equations of motion were integrated with a 2 fs time step. The simulation systems were relaxed by energy minimization and 4 ns of position-restrained MD in the NVT ensemble before starting the production simulations in the NPT ensemble at 25°C and 1 bar for 10 ns, 20 ns or 200 ns. The Bussi-Donadio-Parinello thermostat (*Bussi et al., 2007*), with a $\tau_T$ of 0.1 ps, and the Parinello-Rahman barostat (*Parrinello and Rahman, 1981*), with a $\tau_P$ of 0.5 ps and a compressibility of $4.5 \times 10^5$ bar$^{-1}$, were used.

## Analysis of the MD simulations

The Visual Molecular Dynamics (VMD) program (*Humphrey et al., 1996*), GROMACS (*Van Der Spoel et al., 2005*) and the MDAnalysis Python library (*Michaud-Agrawal et al., 2011*) were used to measure parameters including distances, torsion angles, accessibility and hydrogen bonds from the MD simulations. To validate our MD simulations, amide order parameters ($S^2$) and residual dipolar couplings (RDCs) were calculated from them. Before calculating $S^2$ and RDCs, we removed the overall tumbling from the simulations by aligning each frame to a reference structure. Order parameters were calculated in five ns blocks from the MD trajectories (*Chandrasekhar et al., 1992*) with $S^2$ given by:

$$S^2 = \frac{1}{2}\left[3\sum_{\alpha=1}^{3}\sum_{\beta=1}^{3}\langle\hat{\mu}_\alpha\hat{\mu}_\beta\rangle - 1\right],\tag{1}$$

where α and β denote the x,y,z components of the bond vector μ.

To calculate RDCs, the principle axes of the reference structure were aligned with the experimentally determined alignment tensor. The use of a single reference structure is justified given the stability of the overall fold of nDsbD in the simulations. The RDC for a given residue in a given structure was then calculated using:

$$\bar{D} = \frac{k}{R^3}(A_x \cos^2\theta_1 + A_y \cos^2\theta_2 + A_z \cos^2\theta_3),\tag{2}$$

where κ depends on the gyromagnetic ratios $\gamma_I$ $\gamma_S$, the magnetic permittivity of vacuum $\mu_0$ and Planck's constant and is defined as:

$$k = -\frac{3}{8\pi^2}\gamma_I\gamma_S\mu_0\hbar$$

$R$ is the bond length and was uniformly set to 1.04 Å. The angles $\theta_1$, $\theta_2$ and $\theta_3$ describe the orientation of the individual bonds with respect to the three principal axes of the alignment tensor. $A_x$, $A_y$ and $A_z$ are the principal components of the alignment tensor. Errors in these $S^2$ and RDC values were estimated using random sampling with replacement of the 200 five ns simulation segments.

## Bioinformatics

2753 bacterial and archaeal representative proteomes from the NCBI database were searched for homologs of the N-terminal domain of *Escherichia coli* DsbD (nDsbD) (residues M1-V132) using blastp 2.2.28+ (NCBI) with the following cut-off values: evalue <0.01, identity >20%, coverage >50%. Hits were subsequently used as a blastp query against the *E. coli* MG1655 proteome (evalue <0.01) and only the ones giving DsbD as best hit were retained. A full-length DsbD phylogenetic tree was built using fasttree 2.1.7 (*Price et al., 2010*) using the wag option. A subset of 731 sequences clustering around the *E. coli* DsbD were selected for subsequent analysis as a consistent phylogenetic group of proteins, mostly belonging to the β- and γ-proteobacteria. One sequence per bacterial genus was selected; this gave a set of 134 sequences which were aligned using muscle 3.8.31 (*Edgar, 2004*). Percent conservation of individual positions was calculated using an in-house script.

## Crystallization, data collection, and structure determination of Δloop-nDsbD$_{ox}$

Crystals of Δloop-nDsbD$_{ox}$ were obtained in vapour diffusion hanging drops containing a 1:1 mixture of Δloop-nDsbD$_{ox}$ (10 mg/ml) in 20 mM Tris-HCl (pH 7.5), 50 mM NaCl and reservoir solution containing 28% w/v PEG 4000, 0.1 M ammonium sulphate and 0.1 M sodium acetate (pH 5.0); these conditions had previously yielded diffraction-quality crystals for nDsbD$_{red}$ (*Mavridou et al., 2011*). The reservoir was sealed with vacuum grease and incubated at 16˚C. Crystals were cryoprotected by soaking for a few seconds in 20% v/v glycerol and 80% v/v reservoir solution, and frozen directly in the nitrogen stream of the X-ray apparatus. Datasets were collected with an oscillation angle of 162˚ (1˚/frame) on a Rigaku RU-H3R rotating anode X-ray generator equipped with an R-AXIS IV image plate detector and an Oxford Cryosystems cryostream.

Diffraction data were processed and scaled with the Denzo/Scalepack package (*Otwinowski and Minor, 1997*). Although datasets were initially collected to a resolution of 2.20 Å, completeness was low due to damage of the crystals because of ice formation. It was, therefore, decided to refine the data to 2.60 Å resolution to have satisfactory completeness (overall above 80%). The crystals belonged to space group P2$_1$ ($a$ = 37.33 Å, $b$ = 81.34 Å, $c$ = 46.39 Å and β = 100.66˚). Five percent of reflections were flagged for R$_{free}$ calculations. The structure was solved by molecular replacement with MOLREP (*Murshudov et al., 1997*) using the wild-type nDsbD$_{red}$ structure (PDB accession code 3PFU) as a search model (*Mavridou et al., 2011*), and refined with Phenix.refine (*Adams et al., 2010*) and Coot (*Emsley et al., 2010*), which was used for manual fitting. The refinement converged at R = 25.2% and R$_{free}$ = 29.2%. Two protein molecules were found per asymmetric unit and 59 water molecules were also modelled. No electron density was observed before residue 10 and after residue 120, and before residue nine and after residue 121 in chains A and B, respectively. The coordinates are deposited in the Protein Data Bank (PDB accession code 5NHI).

## Acknowledgements

LSS acknowledges the Biotechnology and Biological Sciences Research Council (BBSRC) for a graduate studentship. DG acknowledges the Swiss National Science Foundation for the Postdoc Mobility and Ambizione Fellowships. MSPS acknowledges funding from the Wellcome Trust and the BBSRC. CR and SJF acknowledge funding from the Wellcome Trust. We thank Prof. D Flemming Hansen for helpful discussions, Nick Soffe for help with the NMR pulse sequence programming, Prof. LE Kay for providing Bruker pulse programs for sign determination, and Albert Magnell for assistance with the expression and purification of Δloop-nDsbD. We are also grateful to all scientists involved in the review of this work for their constructive criticism. The authors acknowledge the use of the University of Oxford Advanced Research Computing (ARC) facility in carrying out this work.

## Additional information

### Funding

| Funder | Grant reference number | Author |
| --- | --- | --- |
| Biotechnology and Biological Sciences Research Council | BB/F01709X/1 | Lukas S Stelzl |
| Biotechnology and Biological Sciences Research Council | BB/R00126X/1 | Mark SP Sansom |
| Wellcome | 208361/Z/17/Z | Mark SP Sansom |
| Wellcome | 079440 | Christina Redfield |
| Wellcome | 092532/Z/10/Z | Stuart J Ferguson Christina Redfield |
| Swiss National Science Foundation | P300PA_167703 | Diego Gonzalez |
| Swiss National Science Foundation | PZ00P3_180142 | Diego Gonzalez |

The funders had no role in study design, data collection and interpretation, or the decision to submit the work for publication.

### Author contributions

Lukas S Stelzl, Conceptualization, Software, Formal analysis, Investigation, Visualization, Methodology, Writing - original draft, Writing - review and editing; Despoina AI Mavridou, Conceptualization, Supervision, Investigation, Visualization, Methodology, Writing - original draft, Writing - review and editing; Emmanuel Saridakis, Diego Gonzalez, Investigation, Writing - review and editing; Andrew J Baldwin, Software, Formal analysis, Visualization, Methodology, Writing - review and editing; Stuart J Ferguson, Conceptualization, Funding acquisition, Writing - review and editing; Mark SP Sansom, Conceptualization, Supervision, Funding acquisition, Methodology, Project administration, Writing - review and editing; Christina Redfield, Conceptualization, Software, Formal analysis, Supervision, Funding acquisition, Investigation, Visualization, Methodology, Writing - original draft, Project administration, Writing - review and editing

### Author ORCIDs

Lukas S Stelzl (iD) https://orcid.org/0000-0002-5348-0277
Despoina AI Mavridou (iD) https://orcid.org/0000-0002-7449-1151
Andrew J Baldwin (iD) http://orcid.org/0000-0001-7579-8844
Mark SP Sansom (iD) https://orcid.org/0000-0001-6360-7959
Christina Redfield (iD) https://orcid.org/0000-0001-7297-7708

### Decision letter and Author response

Decision letter https://doi.org/10.7554/eLife.54661.sa1
Author response https://doi.org/10.7554/eLife.54661.sa2

## Additional files

### Supplementary files
• Transparent reporting form

### Data availability

All data generated or analysed during this study are included in the manuscript and supporting files. Research materials supporting this publication can be accessed by contacting christina.redfield@bioch.ox.ac.uk.

The following dataset was generated:

| Author(s) | Year | Dataset title | Dataset URL | Database and Identifier |
|---|---|---|---|---|
| Saridakis E, Stelzl LS, Mavridou DAI, Redfield C | 2018 | Crystal structure of the E. coli N-terminal domain of DsbD (nDsbD) without the cap-loop region | http://www.rcsb.org/structure/5NHI | RCSB Protein Data Bank, 5NHI |

The following previously published datasets were used:

| Author(s) | Year | Dataset title | Dataset URL | Database and Identifier |
|---|---|---|---|---|
| Mavridou DA, Stelzl L, Ferguson SJ, Redfield C | 2011 | 1H, 13C and 15N resonance assignments for the oxidized state of the N-terminal domain of DsbD from Escherichia coli | http://www.bmrb.wisc.edu/data_library/summary/index.php?bmrbId=17831 | Biological Magnetic Resonance Data Bank, 17831 |
| Mavridou DA, Stelzl L, Ferguson SJ, Redfield C | 2011 | 1H, 13C and 15N resonance assignments for the reduced state of the N-terminal domain of DsbD from Escherichia coli | http://www.bmrb.wisc.edu/data_library/summary/index.php?bmrbId=17830 | Biological Magnetic Resonance Data Bank, 17830 |
| Mavridou DAI, Saridakis E, Ferguson SJ, Redfield C | 2011 | N-terminal domain of Thiol: disulfide interchange protein DsbD in its reduced form | https://www.rcsb.org/structure/3PFU | RCSB Protein Data Bank, 3PFU |

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
