## [Decision Letter]

**Acceptance summary:**

The reviewers were pleased with the revisions which address our concerns and judge that the present work is a nice example of the synergy between NMR and MD simulations.

**Decision letter after peer review:**

Thank you for submitting your article "Local frustration determines loop opening during the catalytic cycle of an oxidoreductase" for consideration by *eLife*. Your article has been reviewed by three peer reviewers, one of whom is a member of our Board of Reviewing Editors, and the evaluation has been overseen by Michael Marletta as the Senior Editor.

There is general enthusiasm for this work and agreement that it is carefully done, providing a nice example of the synergy between solution NMR and MD. Combining these two modalities to elucidate how frustration influences dynamics that in turn influences function is of great interest, and the fact that differences in levels of frustration could potentially affect the mechanism of interaction of the DsbD oxidoreductase, depending on the redox state is fascinating.

After careful consultation between all reviewers we have decided that a revised manuscript should focus on developing the mechanism by which frustration leads to loop opening. The authors have detailed MD simulations of what happens in frustrated (oxidized) and much less frustrated (reduced) systems and trajectories that lead to loop opening. But what is missing is an atomic description of how this is accomplished that careful analysis of your trajectories should supply. In addition, is frustration only characterized by the sidechain of F70 or are other residues involved – that is, could there be a path of frustration which is important for triggering loop opening? Or, is it enough that only 1 residue is involved principally? We list below a number of MAJOR questions to be addressed, but these all reflect a mechanistic understanding that is discussed above.

Essential revisions:

1) What are the "knock-on" effects of the F70 frustration on neighboring side-chains and backbone conformation? Does the presence of the t or g- conformation have any influence on rotamer distributions of neighboring side-chains in the MD simulations? Are the neighboring side-chains packed any differently in the different unit-cells of the crystal structures of Neisseria meningitides shown in Figure 4—figure supplement 2 (the unit cells with either F70 rotamer, and the open conformation)? What about in IVRS X-ray structure? Are the neighboring side-chains behaving differently in the MD simulations that adopt a semi-open conformation?

2) Is it just 1 trajectory that shows the quasi-open state transition? If it is, can authors point to the differences in this trajectory that stabilize this state relative to the other trajectories?

3) Based on the above analyses do any of the conformations of the other side-chains seem important for loop opening? Can the authors postulate on a mechanism for opening?

4) Which conformation of F70 do they authors hypothesize is more likely to result in loop opening? Are there any other concerted structural changes that they think are important?

5) I think it can be a bit misleading to pool the statistics of 5 independent trajectories into a single figure. I would like to see all of the individual distributions from different trajectories (the histograms from Figure 3 and Figure 4) in supplementary figures so it’s more clear if we're observing transitions between states, or observing 1 of the 2 states in each trajectory.

---

## [Author Response]

[…] After careful consultation between all reviewers we have decided that a revised manuscript should focus on developing the mechanism by which frustration leads to loop opening. The authors have detailed MD simulations of what happens in frustrated (oxidized) and much less frustrated (reduced) systems and trajectories that lead to loop opening. But what is missing is an atomic description of how this is accomplished that careful analysis of your trajectories should supply. In addition, is frustration only characterized by the sidechain of F70 or are other residues involved – that is, could there be a path of frustration which is important for triggering loop opening? Or, is it enough that only 1 residue is involved principally? We list below a number of MAJOR questions to be addressed, but these all reflect a mechanistic understanding that is discussed above.

Based on the Reviewing Editors’ summary of the concerns raised by reviewers, the main question we were asked to address is: What is the mechanism by which frustration leads to loop opening? To this end, please find below an explanation of the general approach we have taken in order to analyze our MD trajectories and address this question, as well as our detailed replies to the specific points raised by the reviewers.

As the assessment of the reviewers was generally positive, we have maintained the overall content and layout of our manuscript and, where appropriate, made changes in order to address any concerns raised. To address the main question of how/why the loop opens, we have added a substantial new section in the Results and Discussion entitled ‘Structural features of cap loop opening’ in which we present an in-depth analysis of the MD simulations and discuss the factors that contribute to cap loop opening. In the same section, we provide more characterization of nDsbD when the cap loop is open, and assess the drivers of cap loop closure. To support this new section, we have added one new main figure (Figure 5) and nine figure supplements. Our changes are highlighted in red in the file submitted as a ‘related manuscript file’. Furthermore, we have added two new authors to the manuscript that have provided additional data required to address the questions raised by the reviewers; Diego Gonzalez performed bioinformatics analysis and Emmanuel Saridakis carried out X-ray crystallography, both of which are now included in our revised manuscript. In addition, according to the journal’s instructions, we have added a Key Resources Table and converted our supplementary figures and tables into the *eLife* ‘figure supplement’ and ‘source data’ format.

To identify the determinants of cap loop motion, we have analyzed the MD simulation trajectories in detail and measured distances, torsion angles and hydrogen bonds between key nDsbD amino acids, as well as solvent accessibilities in different redox states of the protein. Our original manuscript showed the F70 ring to C109 Sγdistance, an indicator of cap loop opening, only for selected trajectories (in Figure 3), and presented a summary of overall distance distributions and overall distributions of the F70 χ1 angle in Figures 3D/H and 4B/C. For greater transparency, we have now included plots of the F70 ring to C109 Sγdistance and the F70 χ1 angle for the full 1μs trajectories for both oxidation states (Figure 3—figure supplement 2 and Figure 4—figure supplement 1). This is discussed in greater detail in response to Major Comment (5) below. We have also included data on the solvent accessibility of C103 and C109 Sγ in a 200 ns MD trajectory both for oxidized and reduced nDsbD (Figure 3—figure supplement 1), as further evidence that opening of the cap loop increases the exposure of the cysteines, which in the reduced state remain shielded. We have made minor changes to the text in the Results and Discussion to discuss these new data and figure supplements.

We analyzed the side chain positions in the closed *N. meningitidis* nDsbD_ox_ X-ray structures and in the MD simulations of *E. coli* nDsbD_ox_ showing closed conformations, and found that in addition to alternative conformations for F66/F70, which we previously identified as local structural frustration, the side chain of F67/Y71 also adopts variable conformations, whilst other side chain and backbone positions are not affected. Therefore, the effects of structural frustration at F70 are quite localized in nDsbD but do affect Y71, something we had not reported in our original manuscript. It is interesting to note that our analysis shows that the χ1 torsion angles of F70 and Y71 are not correlated, which indicates that Y71 does not show the frustration identified for F70. This is discussed in more detail in the response to Major Comment (1) below.

We also analyzed the torsion angles of F70 and Y71, their positions relative to each other and to C109 in closed conformations prior to loop opening. We have added a new figure (Figure 5) which shows some of these parameters plotted from 65-105 ns of the major loop opening event (Figure 3F) and support this analysis with more detailed plots of additional parameters for the longer period of 60-200 ns in Figure 5—figure supplements 1, 2 and 3. The plots are complemented by structure ‘snapshots’ taken at various time points during the MD trajectory which help illustrate, in a way that is more accessible to non-MD specialists, how the structure in the vicinity of the cap loop and active site is changing prior to loop opening (Figure 5 and Figure 5—figure supplement 5). Factors leading to loop opening have also been investigated in a further ~20 opening events of duration between 1 and 22 ns. To demonstrate that observations made for the major loop opening are not unique, we have also included a similar detailed analysis of a 22 ns loop opening event in Figure 5—figure supplement 4 and corresponding structural snapshots in Figure 5—figure supplement 5.

Our in-depth analysis shows that loop opening is not initiated from a single, specific conformation of the cap loop and active site. We find that opening can occur from either of the two frustrated conformers of F70, and can also happen concurrently with a change in the F70 χ1. Y71 is also found to adopt both the gauche- and trans χ1 conformations at the time of loop opening (see for example Figure 5—figure supplements 2 and 4). Prior to loop opening, F70 and Y71 sample a number of different conformations. This flexible and dynamic nature of the side chains of F70 and Y71 in the closed state is reflected in their RMSD observed during the MD trajectories which are higher for nDsbD_ox_ than for nDsbD_red_. We propose that the existence of local structural frustration for F70 results in increased dynamics for both F70 and Y71, which leads to occasional loop opening in nDsbD_ox_. This is discussed in more detail in the response to Major Comment (3) below.

We have also analyzed torsion angles, distances and hydrogen bonding after cap loop opening in order to gain a better understanding of the types of conformations that are sampled and are important for full loop opening. After opening the loop can adopt a range of more and less open conformations; these are discussed in the subsection “Structural features of cap loop opening”and illustrated in Figure 5 and its figure supplements. We have extended our analysis to the backbone torsion angles of cap loop residues in the closed and open states of the loop. Clear changes upon loop opening are observed for the φ and ψ values of D68, E69, F70, Y71 and G72 (Figure 5—figure supplement 3). These are correlated with the loss of a hydrogen bond between the Y71 HN and D68 Oδ; this is discussed in the subsection “Structural features of cap loop opening”. Finally, by performing a careful analysis of the MD trajectories we discovered another potentially important role for Y71 coming into play after loop opening. Y71 is found occasionally to adopt ‘blocking’ conformations in which is sits between F70 and C109 and prevents the loop from closing; this would prevent rapid loop closure and might facilitate the interaction of nDsbd_ox_ with cDsbD_red_. In the light of the important roles played by both F70 and Y71, we investigated the conservation of these residues using bioinformatics (Figure 5—figure supplement 6). This analysis shows a high level of conservation of both these residues, and in particular Y71.

Overall, we have analyzed a large number of distances and torsion angles in order to identify the structural determinants of cap loop opening. An alternative could have been to adopt a statistical data analysis approach (Peters and Trout, J. Chem. Phys, 2006, Vreede et al. 2010, PNAS, Okazaki et al. Nat Commun. 2019) or a machine learning approach (Jung et al. ArXiv 2019) in order to extract mechanisms from the MD trajectories. However, these approaches typically require a very large number of transition events (>> 50) to identify such features from the data alone. Loop opening in nDsbD_ox_ is a complex process; it is not a simple two-state transition from a single closed conformation to a single open one. Instead, it can occur from a number of different closed conformations in which the side chains of F70 and Y71 can adopt numerous orientations. The open state is also conformationally highly diverse, with the loop opening to different extents and loop closing occurring from several different open conformations. Some opening events observed in our 1μs trajectories are very transient (< 1 ns), while others are much longer lived (22 ns and > 120 ns). In our analysis, we considered ~20 different spontaneous opening events. This is not a large enough number of openings to apply the statistical data analysis or machine learning approaches. MD simulations run for much longer than the 1μs reported here would be required in order to adopt this approach; this is well beyond the scope of the current study, especially in the light of the constraints imposed by the current global pandemic. Nonetheless, we are confident that the rigorous descriptive and structurally based approach we have adopted is relevant here. It has unveiled consistent trends and has provided important insights into the mechanism of loop opening, which along with our previously submitted analysis of frustration significantly advance our understanding of the mechanism of reductant transfer in nDsbD. Importantly, prior structural knowledge, as we present here, can greatly aid in any future data driven approaches (Okazaki et al.Nat Commun.2019) and can be critical to assess the results from purely automatic approaches.

Essential revisions:1) What are the "knock-on" effects of the F70 frustration on neighboring side-chains and backbone conformation? Does the presence of the t or g- conformation have any influence on rotamer distributions of neighboring side-chains in the MD simulations? Are the neighboring side-chains packed any differently in the different unit-cells of the crystal structures of Neisseria meningitides shown in Figure 4—figure supplement 2 (the unit cells with either F70 rotamer, and the open conformation)? What about in IVRS X-ray structure? Are the neighboring side-chains behaving differently in the MD simulations that adopt a semi-open conformation?

We have performed further analysis of the X-ray structure of *N. meningitidis* nDsbD and found that the alternative ‘frustrated’ conformations observed for the side chain of F66 (homologous to F70) were accompanied by variable positions for the side chain of F67 (homologous to Y71); the positions of both these residues are now illustrated in Figure 4—figure supplement 2. No other side-chain rotamer distributions are perturbed in *N. meningitidis* nDsbD by the alternative positions for F66, nor is the backbone significantly perturbed.

This observation for *N. meningitidis* nDsbD indicated that we should look at the behaviour of the side chain of Y71 (the residue homologous to F66) in our MD simulations. Like F70, Y71 also samples both gauche- and trans χ1 conformers. However, unlike F70 which shows an almost equal distribution between the two χ1 conformations in the MD simulations, Y71 shows a preference for the gauche- conformation (86% in closed structures, 78% in open structures). Interestingly, we observe that there is no correlation between the χ1 values of F70 and Y71; the less common Y71 trans conformer is observed for both frustrated F70 conformers. This is discussed in the subsection “Structural features of cap loop opening” and the lack of correlation in the χ1 values of F70 and Y71 is illustrated in the plots shown in Figure 5—figure supplement 2 and Figure 5—figure supplement 4. As in *N. meningitidis* nDsbD, we also found that in the closed structures of *E. coli* nDsbD_ox_ no other side-chains are perturbed by the sampling of gauche- and trans rotamers by F70 and Y71.

The reviewer asks about side-chain packing in the crystal structure of oxidized *N. meningitidis* nDsbD. The side chains of F66/F67 are found at the surface of the nDsbD structure and in proximity to F66/F67 of other molecules in the unit cell and in neighbouring unit cells. F66/F67 of molecules A and B are in close proximity to each other within the unit cell, while F66/F67 in molecule C are close to F66/F67 in molecule F in a neighbouring unit cell; these 4 molecules all show a closed loop conformation. F66/F67 in molecule E, which is the only molecule with an open loop conformation, are close to F66/F67 in molecule D, which has a closed loop, in a neighbouring unit cell. Although the crystal contacts may stabilize the F66/F67 side chain positions, the striking similarity between these conformations and the conformations found for F70/Y71 in our MD simulations strongly suggests that *N. meningitidis* nDsbD does sample these conformations in solution and that a variety of them got ‘trapped’ during the crystallization process. This conclusion is further supported by the fact that only a single closed conformation is observed in the crystal structure of reduced *N. meningitidis* nDsbD. In the *E. coli* 1VRS structure, the nDsbD cap loop is in an open conformation not because of crystal contacts but because nDsbD is in complex with cDsbD; the same loop conformation is observed for both complexes in the unit cell.

2) Is it just 1 trajectory that shows the quasi-open state transition? If it is, can authors point to the differences in this trajectory that stabilize this state relative to the other trajectories?

The ~120 ns cap loop opening highlighted in Figure 3F is the major loop opening event we observed in the 1 μs MD simulations of nDsbD_ox_, but it is not the only opening event. This was not clear in our original manuscript and we have clarified this in the revised version. We have added a figure supplement to Figure 3 (Figure 3—figure supplement 2) in which we show the F70-C109 distance in the 600 ns of trajectories not shown in Figure 3 (for both oxidized and reduced states). Very short-lived loop openings such as those observed in Figure 3E are also observed in these additional trajectories for nDsbD_ox_, along with ~20 loop opening events with a duration of at least 1 ns and a F70-C109 distance of at least 10 Å; this is discussed in the Results and Discussion. The longest of these additional opening events (~22 ns shown in the middle right-hand panel of Figure 3—figure supplement 2) is analyzed in more detail in Figure 5—figure supplements 4 and 5.

3) Based on the above analyses do any of the conformations of the other side-chains seem important for loop opening? Can the authors postulate on a mechanism for opening?

As explained above in Major Comment (1), we find that Y71, like F70, samples both gauche- and trans χ1 conformers in the MD simulations, but the values for the two residues are not correlated. As discussed in Major Comment (4) below, loop opening does not occur from only one of the frustrated F70 χ1 conformers. Instead, we find that the increased dynamics of F70 and Y71 and the relative orientation of F70 and Y71 to each other and to C109 are important for cap loop opening. The sequence of events leading to the major loop opening shown in Figure 3F is analyzed in depth in Figure 5 and Figure 5—figure supplements 1, 2, 3. More specifically, although the cap loop remains closed from 60-80 ns, the side chains of both F70 and Y71 sample a number of different conformations. Uncorrelated changes in their side chain torsion angles can occasionally bring the F70 and Y71 rings closer together than observed in the 1L6P X-ray structure, whilst at other times the ring of Y71 transiently moves into more exposed conformations. This flexible and dynamic nature of the side chains of F70 and Y71 in the closed state is reflected in their RMSD during the MD trajectories, compared to the equilibrated X-ray structures. In the nDsbD_ox_ 1 μs trajectories, F70 and Y71 have RMSDs of 2.8 ± 0.8 and 2.6 ± 0.7 Å, respectively when the loop is closed (F70-C109 < 6 Å). By contrast, these RMSD values are 1.7 ± 0.6 and 1.6 ± 0.8 for F70 and Y71, respectively in nDsbD_red_ trajectories. Taking these observations together, we propose that the existence of local structural frustration for F70 results in increased dynamics for both F70 and Y71, which leads to occasional loop opening in nDsbD_ox_. In addition, by analyzing the open parts of the MD trajectories and trajectories starting from an open loop conformation, we find another important role for Y71. This residue is found occasionally to adopt ‘blocking’ orientations in which is sits between F70 and C109 and prevents the loop from closing; this may prevent rapid loop closure and facilitate the interaction of nDsbd_ox_ with cDsbD_red_. In the light of the important roles played by F70 and Y71, we investigated the conservation of these residues using bioinformatics (Figure 5—figure supplement 6). This analysis shows a high level of conservation of both these residues (especially Y71) suggesting that our observations are relevant across many bacterial species. Overall, the roles of F70 and Y71 in cap loop motion are discussed in the subsection “Structural features of cap loop opening”.

4) Which conformation of F70 do they authors hypothesize is more likely to result in loop opening? Are there any other concerted structural changes that they think are important?

We investigated whether cap loop opening is linked directly to one of the two frustrated F70 χ1 conformers, something that would be the most natural assumption. The major cap loop opening of nDsbD_ox_ at 82.5 ns in Figure 3F coincides with a change of the F70 χ1 from gauche- to trans. To study this further, we focused on several shorter cap loop opening events occurring during the 1 μs trajectory of nDsbD_ox_. We observed twenty events with a duration of at least 1 ns and a F70-to-C109 distance of at least 10 Å, which we used to assess the F70 conformation at the point of loop opening. In only one of these events does F70 undergo a similar gauche- to trans transition at the point of opening. Fourteen of the openings start from F70 in the trans conformer, while five openings start from the gauche- conformer. Therefore, loop opening is not coupled to a specific F70 χ1 conformer, although more loop openings occur from the trans conformer. This is discussed in the subsection “Structural features of cap loop opening”.

5) I think it can be a bit misleading to pool the statistics of 5 independent trajectories into a single figure. I would like to see all of the individual distributions from different trajectories (The histograms from Figure 3 and Figure 4) in supplementary figures so it’s more clear if we're observing transitions between states, or observing 1 of the 2 states in each trajectory.

We thank the reviewer for making this very important point. Here we pool the statistics of 14 independent trajectories starting from the same starting structure, and differing only in their initial velocities. In this case, pooling the statistics from independent trajectories is permissible, since running multiple simulation trajectories from the same structure can give superior results compared to running a single long trajectory (Knapp et al., JCTC, 2018). More generally, in the case of independent simulations started from different structures, pooling is of course only possible with further modeling, e.g., employing a Markov-state model (Swope et al., J. Phys Chem. B, 2004, Stelzl and Hummer, JCTC 2017, Stelzl et al., JCTC 2017) or a diffusion model (Hummer, New J. Phys., 2005).

We agree with the reviewer that the individual distributions of the different simulation trajectories are potentially informative and, thus have added figure supplements that complement Figures 3 and 4. In Figure 3—figure supplement 2, we show the F70-C109 distance for the additional 600 ns of trajectories that are not shown in Figure 3 (for both reduced and oxidized nDsbD). These give a clear illustration of the distribution of closed and open loop structures (as assessed by the F70-C109 distance) that is summarized in Figure 3D/H. Figure 3 and this figure supplement illustrate further that the cap loop of oxidized nDsbD is highly dynamic in multiple different trajectories and opens many times (see response (2) above), while reduced nDsbD features a rigid closed cap loop.

Further, Figure 4—figure supplement 1 shows the F70 χ1 angle plotted for the full 1μs trajectories of both reduced and oxidized nDsbD; the distributions in Figure 4B/E are based on these data. This new figure shows that the F70 side chain interconverts frequently between the gauche- and trans conformations in oxidized nDsbD. By comparing this supplement to Figure 3 and Figure 3—figure supplement 2, it becomes clear that these interconversions of F70 χ1 occur in both the closed and open conformations. In the legend of this new figure supplement we state the % of the gauche- and trans conformers in closed versus open structures for oxidized nDsbD. It can also be seen that virtually no transitions are observed for reduced nDsbD (the exception is briefly at ~100 ns in the top panel).

Finally, the new figure, Figure 5, and several of its supplements (Figure 5—figure supplements 1-5) provide detailed distance and angle information for the two longest (~120 ns and ~20 ns) loop opening events.